# Egocentric Planning for Scalable Embodied Task Achievement

**Xiaotian Liu**[*]
ServiceNow Research
Montreal, QC, Canada
`xiaotian.liu`
`@mail.utoronto.ca`

**Hector Palacios**[†]
ServiceNow Research
Montreal, QC, Canada
`hectorpal`
`@gmail.com`

**Christian Muise**
Queen's University
Kingston, ON, Canada
`christian.muise`
`@queensu.ca`

## Abstract

Embodied agents face significant challenges when tasked with performing actions in diverse environments, particularly in generalizing across object types and executing suitable actions to accomplish tasks. Furthermore, agents should exhibit robustness, minimizing the execution of illegal actions. In this work, we present Egocentric Planning, an innovative approach that combines symbolic planning and Object-oriented POMDPs to solve tasks in complex environments, harnessing existing models for visual perception and natural language processing. We evaluated our approach in ALFRED, a simulated environment designed for domestic tasks, and demonstrated its high scalability, achieving an impressive 36.07% unseen success rate in the ALFRED benchmark and winning the ALFRED challenge at CVPR Embodied AI workshop. Our method requires reliable perception and the specification or learning of a symbolic description of the preconditions and effects of the agent's actions, as well as what object types reveal information about others. It can naturally scale to solve new tasks beyond ALFRED, as long as they can be solved using the available skills. This work offers a solid baseline for studying end-to-end and hybrid methods that aim to generalize to new tasks, including recent approaches relying on LLMs, but often struggle to scale to long sequences of actions or produce robust plans for novel tasks.

## 1 Introduction

Embodied task accomplishment requires an agent to process multi-modal information and plan over long task horizons. Recent advancements in deep learning (DL) models have made grounding visual and natural language information faster and more reliable (MBP[+]21). As a result, embodied task-oriented agents have been the subject of growing interest (STG[+]20, YRT[+]22, WDKM21). Benchmarks such as the *Action Learning From Realistic Environments and Directives* (ALFRED) were proposed to test embodied agents' ability to act in an unknown environment and follow language instructions or task descriptions (STG[+]20). The success of DL has led researchers to attempt end-to-end neural methods (ZC21, SGT[+]21). In an environment like ALFRED, these methods are mostly framed as imitation learning, where neural networks are trained via expert trajectories. However, end-to-end optimization leads to entangled latent state representation where compositional and long-horizon tasks are challenging to solve. Other approaches use neural networks to ground visual information into persistent memory structures to store information (MCR[+]22, BPF[+]21). These approaches rely on templates of existing tasks, making them difficult to generalize to new problems or unexpected action outcomes. ALFRED agents must navigate long horizons and skill assembly, operating within a deterministic environment that only changes due to the agent's actions.

---

[*]Work done when Xiaotian Liu was an intern at ServiceNow Research.
[†]Work done when Hector Palacios was a research scientist at ServiceNow Research.

37th Conference on Neural Information Processing Systems (NeurIPS 2023).

Long sequential decision problems with sparse rewards are notoriously difficult to train for gradient-based reinforcement learning (RL) agents. But a symbolic planner with a well-defined domain can produce action sequences composed of hundreds of actions in less than a fraction of a second for many tasks. Embodied agents might need to conduct high-level reasoning in domains with long action sequences. In such scenarios, human experts can opt to model the task within a lifted planning language. Planning languages such as *Planning Domain Definition Language*, or PDDL (HLMM19), is the most natural candidate, given the high scalability of planners that use such languages (GB13). For instance, we define the tasks of a home cooking robot with actions, such as *opening*, *move-to*, *pick-up*, *put-on*, and objects, such as *desk*, *stove*, *egg* found in common recipes.

However, most symbolic planning techniques are relatively slow in partially observable environments with a rich set of objects and actions. Thus we propose an online iterative approach that allows our agent to switch between exploration and plan execution. We explicitly model the process of actively gathering missing information by defining a set of *unknown-objects* and *exploration-actions* that reveals relevant information about an agent's goal. Instead of planning for possible contingencies, we use an off-the-shelf classical planner(Hof01) for fully-observable environments to determine whether the current state has enough knowledge to achieve the goal. If more information is needed, our agent will switch to a goal-oriented exploration to gather missing information. Our approach can seamlessly incorporate any exploration heuristics, whether they are learned or specified.

We implemented and evaluated our method on the popular embodied benchmark ALFRED using only natural language task descriptions. To demonstrate the effectiveness of our method, we use the same neural networks for both visual and language grounding used in FILM, which was the SOTA and winner of the 2021 ALFRED challenge (MCR$^+$22). See a comparison with other methods in Table 1. By replacing FILM's template-based policies with our egocentric iterative planner, our method improved FILM's success rate (SR) by 8.3% (or a 30.0% relative performance increase) in unseen environments winning the the ALFRED challenge at CVPR 2022 Embodied AI workshop. Our empirical results show that the performance increase was attributed to a more robust set of policies that account for goal-oriented exploration and action failures. We also show that our method can conduct zero-shot generalization for new tasks using objects and actions defined in the ALFRED setting.

## 2    ALFRED Challenge

ALFRED is a recognized benchmark for Embodied Instruction Following (EIF), focused on training agents to complete household tasks using natural language descriptions and first-person visual input. Using a predetermined set of actions, the agent operates in a virtual simulator, AI2Thor (KMG$^+$17), to complete a user's task from one of seven specific classes instantiated in concrete objects. For example, a task description may be "Put a clean sponge on the table," and instructions that suggest steps to fulfill the task. The agent must conduct multiple steps, including navigation, interaction with objects, and manipulation of the environment. Agents must complete a task within 1000 steps with fewer than 10 action failures. We base our approach on task descriptions rather than the provided instructions. Task complexity varies, and an episode is deemed successful if all sub-tasks are accomplished within these constraints. Evaluation metrics are outlined in Section 8.

## 3    Overview of Egocentric Planning for ALFRED

Figure 1 illustrates our proposed method integrating several components to facilitate navigation and task completion: a visual module responsible for semantic segmentation and depth estimation, a language module for goal extraction, a semantic spatial graph for scene memorization, and an egocentric planner for planning and inference. Initially, high-level language task description is utilized to extract goal information, and the agent is provided with a random exploration budget of 500 steps to explore its surroundings. The random exploration step is used to generate diverse set of object cluster for further exploration using our planner. Subsequently, at t = 500, the gathered information from the semantic spatial graph is converted into a PDDL problem for the agent. We employ a novel open-loop replanning approach, powered by an off-the-shelf planner, to support further exploration and goal planning effectively.

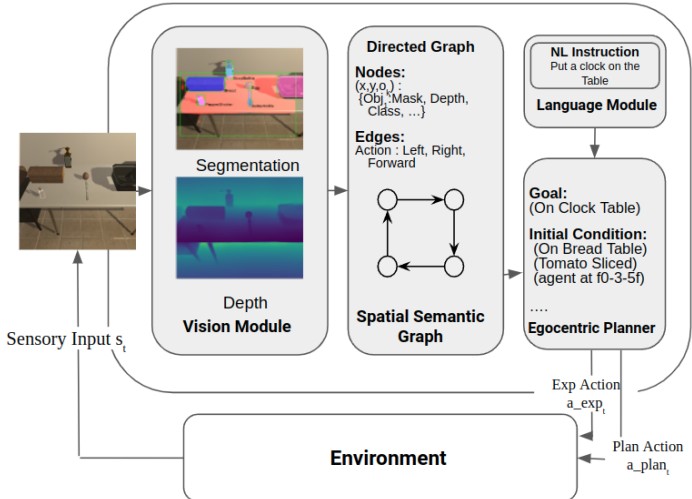

Figure 1: Egocentric Planning for ALFRED

To facilitate egocentric planning, we first define the ontology of the planning problems, which encompasses action schemas, object types, and potential facts schemas. The natural language task description is subsequently converted into a planning goal. Following the initial exploration phase, the semantic spatial graph is updated to indicate the new position of the agent and its perceptions. The algorithm iterates through these steps: a) finding a plan for achieving the goal and returning it upon success; b) if no plan exists, replacing the goal with another fact, called (explore), associated with unvisited states. The semantic spatial graph is updated during each iteration to construct the new initial state of the agent, allowing for an incremental egocentric view of the environment.

## 4 Embodied Agents: Generalization within Object Types and Skills

Embodied Agents are task solvers in environments linked to a hidden Object-oriented POMDP, emphasizing objects, actions/skills, and costs rather than rewards. We assume that these objects mediate perceptions and actions. An *Environment* is a tuple, $\mathcal{E} = \langle A_{\mathcal{E}}, \mathcal{T}_{\mathcal{E}}, \mathcal{V}_{\mathcal{E}}, \mathsf{reset}, \mathsf{step} \rangle$, with $A_{\mathcal{E}}$ being the set of parameterized actions, $\mathcal{T}_{\mathcal{E}}$ defining the set of object types and potential values $\mathcal{V}_{\mathcal{E}}$, reset representing the first observation, and step representing an action execution that returns an observation and cost. A *Task* is represented by a tuple, $T_{\mathcal{E}} = \langle I_{\mathcal{E}}, G_{\mathcal{E}} \rangle$, which sets the initial state and the goal of the environment via the reset function. Lastly, *Embodied Agents* employ functions $M_I$ to initialize their mental state and $M_U$ to update it, have a policy $\pi_M$ to select actions and a function $G_M$ for deciding when they consider the goal to be achieved.

Agents that tackle a stream of tasks within the same environment have incentives to specialize to the known object types and their inherent capabilities expressed through parameterized actions. Such over-specialization provides an opportunity for developing robust agents that can generalize across different tasks within the same environment. This principle applies to physical and software agents whose actions are grounded in these objects. Our framework forms a basis for creating flexible and adaptable agents that can perceive and act upon recognized object types. Further details and implications are covered in Section D.

## 5 Egocentric Planning for Embodied Agents

Here, we introduce *Egocentric Planning*, an approach for embodied agents to solve tasks in Object-oriented POMDP environments. The approach requires a symbolic planning model that assumes full observability for testing both action applicability and goal achievement. However, the complexity of specifying or learning such a symbolic model is simplified by its alignment with the Object-oriented POMDP environment, object types, and action signatures. Egocentric planning relies on a reasonable assumption to derive a sensor model, equivalent to a function that updates the distribution of possible

current states given new observations (APG09). For instance, in a planning domain with objects for different types and colors, the sensor model can say that cubes and rectangles are red, while ovals and circles are black. When the agent examines an object, and observes that it is black, the agent should filter the belief to represent that the object cannot be a cube or a rectangle. In contrast, our method requires a set of *anchor object types* that are assumed to reveal information about other objects and *exploration actions* that allow visiting new anchor objects, revealing new objects and their properties. The method can be combined with standalone models for processing images or natural language. The resulting method is both theoretically sound, assuming that the symbolic planning model is correct, and practically applicable, as it leverages the speed of symbolic planners.

We present an abstract algorithm that leverages the objects types and action signatures of the Object-oriented POMDP to ground the required symbolic planning model.

### 5.1 Background: Parameterized Full-Observable Symbolic Planning

In the supplementary material, Section E, we define *Parameterized Full-Observable Symbolic Planning domains and problems*. A *planning domain* is defined as a tuple $\mathcal{PD} = \langle \mathcal{T}, \mathcal{V}, \mathcal{P}, \mathcal{A}, \Phi \rangle$ where $\mathcal{T}$ is a finite set of object types with $\mathcal{V}$ a finite set of values, $\mathcal{P}$ is a finite set of typed predicates, $\mathcal{A}$ is a finite set of typed actions with preconditions and effects, and $\Phi$ is a function that maps each action and predicates to a tuple expressing their typed arguments. A *planning problem* consists of a planning domain, a finite set of objects, and descriptions of the initial state and the goal.

For ALFRED, we rely on *Classical Planning*, a kind of full-observable planning that assumes deterministic actions, where preconditions are a set of predicates, and effects are a pair of sets of predicates called add and delete, representing the action's modifications. Correspondingly, a parametric classical planning problem is a full-observable planning problem where the initial state and the goal are conjunctions of grounded predicates.

### 5.2 Egocentric Planning for Embodied Agents

Egocentric planning employs a user-specified fully-observable symbolic planning model to address tasks in Object-oriented POMDP environments. In the next section, we detail the specifics of our ALFRED implementation. As actions in ALFRED are deterministic, we can apply a classical planner to solve the associated planning problems. We first present a general algorithm for parametric full-observable planning problems.

---

**Algorithm 1**
Iterative Exploration Replanning (IER)

---

**Input:** Environment $\langle A_{\mathcal{E}}, \mathcal{T}_{\mathcal{E}}, \mathcal{V}_{\mathcal{E}}, \mathsf{reset}, \mathsf{step} \rangle$
**Input:** Planning domain $\mathcal{PD} = \langle \mathcal{T}, \mathcal{V}, \mathcal{P}, \mathcal{A}, \Phi \rangle$
**Input:** Anchor Types & Exploration Acts $\langle \mathcal{T}_a, \mathcal{X} \rangle$
**Input:** Mental state Init & Update $\langle M_I^{\mathcal{PD}}, M_U^{\mathcal{PD}} \rangle$
**Input:** Task $\langle I_{\mathcal{E}}, G_{\mathcal{E}} \rangle$
**Output:** Successful trace $\tau$ or Failure

1: $p \leftarrow \mathsf{reset}(I_{\mathcal{E}}, G_{\mathcal{E}})$     ▷ Initial perception
2: $\mathcal{O}, I, G \leftarrow M_I^{\mathcal{PD}}(p, I_{\mathcal{E}}, G_{\mathcal{E}})$
3: $\mathcal{C} \leftarrow \{o' \mid \mathsf{type}(o') \in \mathcal{T}_a \text{ and } o' \text{ occurs in } I\}$
        ▷ $\mathcal{C}$: Observed Anchor Objects
4: $\tau, \mathsf{solved} \leftarrow [\,], False$
5: **while** not solved **do**
6:      $\pi_{solve} \leftarrow \mathsf{Solve}(\langle \mathcal{PD}, \mathcal{O}, I, G \rangle)$
7:      **if** $\pi_{solve}$ is *None* **then**
8:          $\mathcal{A}_e \leftarrow \mathcal{A}$
9:          **for** $o \in \mathcal{O}$ and $\mathsf{type}(o) \in \mathcal{T}_a$ **do**
10:             **if** $o \notin \mathcal{C}$ **then**
11:               $I \leftarrow I \cup \{(\text{unknown } o)\}$
12:             **if** $(\text{unknown } o) \in I$ **then**
13:               $\mathcal{A}_e \leftarrow \mathcal{A}_e \cup \{\mathit{ExploreAct}(a, o)$
                       $\text{for } a \in \mathcal{X}\}$
14:          $G_e \leftarrow \{(\text{explored})\}$
15:          $\pi_{explore} \leftarrow$
              $\mathsf{Solve}(\langle \mathcal{PD} \text{ with } \mathcal{A}_e, \mathcal{O}, I, G_e \rangle)$
16:          **return** Failure if **if** $\pi_{explore}$ is *None*
17:          **for** $a \in \pi_{explore}$ **do**
18:             $p, c \leftarrow \mathsf{step}(a)$
19:             Break **if** failed $\in p$
20:             $\tau.\mathsf{append}(a)$
21:             $\mathcal{O}, I \leftarrow M_U^{\mathcal{PD}}(\mathcal{O}, I, a, p, c)$
22:             **if** $a \in \mathcal{X}$ **then**
23:               $\mathcal{C} \leftarrow \mathcal{C} \cup \{o' \mid \mathsf{type}(o') \in \mathcal{T}_a$
                 $\text{and } o' \text{ argument of } a\}$
24:      **else**
25:          **for** $a \in \pi_{solve}$ **do**
26:             $p, c \leftarrow \mathsf{step}(a)$
27:             Break **if** failed $\in p$
28:             $\tau.\mathsf{append}(a)$
29:             $\mathcal{O}, I \leftarrow M_U^{\mathcal{PD}}(\mathcal{O}, I, a, p, c)$
30:      $\mathsf{solved} \leftarrow$ True **if** $I$ satisfies $G$
31: **return** $\tau$

---

Algorithm 1 implements our Egocentric Planning approach by constructing a sequence of planning problems, alternating exploration and task solving. At each step, the agent's mental state is updated though sensory input to represent the current state and the known objects that will be part of the

following planning problem. The agent deems the goal achieved when it finds a plan for achieving it and executes the plan successfully. Our policy ($\pi_M$) leverages a standalone planner for both exploration and task solving.

Our exploration and sensing strategy hinges on two elements. First, *anchor object types* $\mathcal{T}_a \subseteq \mathcal{T}$ are types of objects that reveal the presence of others. For example, in ALFRED, *location* is the sole anchor object type. Second, *exploration actions* $\mathcal{X} \subseteq \mathcal{A}$ enables the discovery of previously unknown objects. For instance, moving to a new location in ALFRED enables observing new items and locations. This object-centric view of the environment simplifies the creation of a symbolic model required for planning. The function *ExploreAct* $(a, o)$ returns a copy of the function where (`unknown` $o$) is added to the precondition, and the effect is extended to cause (`explored`) to be true and (unknown $o$)) to be false.

Our agent's *mental state* is represented by a set of objects and an initial state of grounded predicates, $(\mathcal{O}, I)$. The function $M_I^{\mathcal{PD}}$ initializes the symbolic goal $G$ and sets $(\mathcal{O}, I)$ with the initial objects and true grounded predicates. The mental state update function, $M_U^{\mathcal{PD}}$, updates $(\mathcal{O}, I)$ given the executed action $a$, the returned perception $p$, and the cost $c$ of executing $a$. For example, the mental state update in ALFRED adds new objects observed in the perception. It updates the initial state with these new objects and their properties, including their relationship with non-perceived objects like location. Both functions maintain the alignment between the current planning state and the environment with an underlying object-oriented POMDP model.

Specifically, the actions, object types, and the values of the environment may be different from the planning domain's. While the ALFRED environment supports movement actions like `move-forward` or `turn-left`, the planning domain might abstract these into a `move` action. Although the ALFRED environment does not include a `location` type, we manually specify it so movable objects can be associated with a specific location. Indeed, the first initial state $I$ includes a first location and its surrounding ones. Thus, the agent can revisit the location once it has found a solution involving that object. While integrating these components may be challenging, it is a hurdle common to all embodied agent approaches aiming to generalize across components under a constrained training budget. In general, practical applications will likely monitor the agent's performance regarding object types and skills. We discuss the advantages and drawbacks of our approach in Section 9.

# 6 Approach for ALFRED

## 6.1 Model Overview

Our method is structurally similar to FILM but we substitutes the SLAM-style map with a graph structure for object and location information storage (MCR$^+$22). The action controller employs our iterative egocentric planner that utilizes a semantic location graph for task completion and exploration. The node of the graph are uniquely labeled by the coordinates (x-coordinate, y-coordinate, facing direction), and the edges are labeled by actions(turn-right, turn-left, move-forward). Each node in the graph stores object classes, segmentation masks, and depth information generated by the vision module. We label the nodes with "known" if the agent has visited the particular location and "unknown" if the node is only observed. Unknown nodes on the graph are prioritized for exploration based on objects they contain. This setup promotes more robust action sequences and generalization beyond the seven ALFRED-defined tasks. At each timestep, the vision module processes an egocentric image of the environment into a depth map using a U-Net, and object masks using Mask2-RCNN (HGDG17, LCQ$^+$18). The module then computes the average depth of each object and stores only those with an average reachable distance less than 1.5. A confidence score, calculated using the sum of the object masks, assists the egocentric planner in prioritizing object interaction. We employ text classifier to transform high-level language task description into goal conditions. Two separated models determine the task type and the objects and their properties. For instance, a task description like "put an apple on the kitchen table" would result in the identification of a "pick-up-and-place" task and a goal condition of (`on apple table`). We convert the FILM-provided template-based result into goal conditions suitable for our planner. The spatial graph, acting as the agent's memory during exploration, bridges grounded objects and the state representation required for our planner. The graph, updated and expanded by policies produced by our egocentric planner, encodes location as the node key and visual observations as values, with edges representing agent actions.

## 6.2 Egocentric Agent for ALFRED

The object-oriented environment for ALFRED, denoted as $\mathcal{E} = \langle A_{\mathcal{E}}, \mathcal{T}_{\mathcal{E}}, \mathcal{V}_{\mathcal{E}}, \mathsf{reset}, \mathsf{step} \rangle$, includes actions such as pick-up, put-down, toggle, slice, move-forward, turn-left, and turn-right. The FILM vision module is trained as flat recognizer of all possible object types. We postprocess such detection into high level types Object and Receptacle in $\mathcal{T}_{\mathcal{E}}$, with corresponding $\mathcal{V}_{\mathcal{E}}$ indicating subtypes and additional properties. For instance, the object type includes subtypes like apple and bread, while Receptacle includes subtypes such as counter-top and microwave. Values incorporate properties like canHeat, canSlice, isHeated, and isCooled.

The planning domain for ALFRED is denoted $\mathcal{PD} = \langle \mathcal{T}, \mathcal{V}, \mathcal{P}, \mathcal{A} \rangle$. $\mathcal{T}$ includes the Location type, with corresponding values in $\mathcal{V}$ representing inferred locations as actions are deterministic. Predicates in $\mathcal{P}$ represent the connection between locations and the presence of objects at those locations. Further predicates express object properties, such as canHeat, canSlice, isHeated, and isCooled. Unlike the ALFRED environment, actions in the planning domain are parametric and operate on objects and locations. Instead of move-forward, the planning domain features a move action, representing an appropriate combination of move-forward, turn-left, and turn-right. For actions like toggle in ALFRED, which have different effects depending on the object in focus, we introduce high-level, parametric actions like heat that can be instantiated with the relevant object. As the actions in ALFRED are deterministic, we use a classical planner, enabling high scalability by solving one or two fresh planning problem per iteration of Algorithm 1.

# 7 Related Work

Visual Language Navigation (VLN) involves navigating in unknown environments using language input and visual feedback. In this context, we focus on the ALFRED dataset and methods based on the AI2-Thor simulator, which serve as the foundation for various approaches to embodied agent tasks. Initial attempts on ALFRED employed end-to-end methods, such as a Seq2Seq model (STG[+]20), and transformers (ZC21). However, due to the long episodic sequences and limited training data, most top-performing VLN models for ALFRED employ modular setups, incorporating vision and language modules coupled with a higher-level decision-making module (MCR[+]22, MC22, IO22). These methods typically rely on hand-crafted scripts with a predefined policy. However, this approach has several downsides. First, domain experts must manually specify policies for each new task type, regardless of the changes involved. Second, fixed-policy approaches do not allow the agent to recover from errors in the vision and language modules, as the predefined policy remains unchanged.

Recently, there has been growing interest in utilizing planning as a high-level reasoning method for embodied agents. Notable work in this area includes OGAMUS (LSS[+]22), and DANLI (ZYP[+]22). These methods demonstrate the effectiveness of planning in achieving more transparent decision-making compared to end-to-end methods while requiring less domain engineering than handcrafted approaches. However, these works primarily tested in simpler tasks or rely on handcrafted planners specifically tailored to address a subset of embodied agent problems.

Large Language Models (LLMs) have recently been harnessed to guide embodied actions, offering new problem-solving approaches. Works like Ichter et al.'s SayCan utilize LLMs' likelihoods and robots' affordances, grounding instructions within physical limitations and environmental context (iBC[+]23). Song et al.'s (SWW[+]23) LLM-Planner and Yao et al.'s (YZY[+]23) ReAct have further demonstrated the potential of LLMs for planning and reasoning for embodied agents. Alternative methods not using LLMs, include Jia et al. and Bhambri et al.'s hierarchical methods utilizing task instructions(JLZ[+]22, BKMC22). However, the scalability and robustness of these methods is under study. Our approach, in contrast, is ready to capitalize on the expressivity of objects and actions for superior generalization in new tasks, ideal for rich environments with multiple domain constraints. This challenges of using LLM for symbolic planning is noted by Valmeekam et al. (VSM[+]23) who showed the weakness of prompting-based approaches, and Pallagani et al. (PMM[+]22), who highlight the massive amount of data required for reaching high accuracy.

# 8 Experiments and Results

The ALFRED dataset contains a validation dataset which is split into 820 *Validation Seen* episodes and 821 *Validation Unseen* episodes. The difference between *Seen* and *Unseen* is whether the room

|          | Test Seen |       |       |       | Test Unseen |       |       |       |
|----------|-----------|-------|-------|-------|-------------|-------|-------|-------|
|          | SR        | GC    | PLWSR | PLWGC | SR          | GC    | PLWSR | PLWGC |
| Seq2Seq  | 3.98      | 9.42  | 2.02  | 6.27  | 0.39        | 7.03  | 0.08  | 4.26  |
| ET       | 38.42     | 45.44 | 27.78 | 34.93 | 8.57        | 18.56 | 4.1   | 11.46 |
| HLSM     | 25.11     | 35.15 | 10.39 | 14.17 | 24.46       | 34.75 | 9.67  | 13.13 |
| FILM     | 28.83     | 39.55 | 11.27 | 15.59 | 27.8        | 38.52 | 11.32 | 15.13 |
| LGS-RPA  | 40.05     | 48.66 | 21.28 | 28.97 | 35.41       | 45.24 | 15.68 | 22.76 |
| **EPA**  | 39.96     | 44.14 | 2.56  | 3.47  | **36.07**   | 39.54 | 2.92  | 3.91  |
| Prompter | 53.23     | 64.43 | 25.81 | 30.72 | 45.72       | 58.76 | 20.76 | 26.22 |

Table 1: Comparison of our method with other methods on the ALFRED challenge. The challenge declare as winner the method with higher Unseen Success Rate (SR). For all metrics, higher is better. EPA is our approach. Under the line are approaches submitted after the challenge leader board closed.

environment is available in the training set. We use the validation set for the purpose of fine-tuning. The test dataset contains 1533 *Test Seen* episodes and 1529 *Test Unseen* episodes. The labels for the test dataset are contained in an online server and are hidden from the users. Four different metrics are used when evaluating ALFRED results. Success Rate (SR) is used to determine whether the goal is achieved for a particular task. Goal-condition Success is used to evaluate the percentage of subgoal-conditions met during an episode. For example, if an objective is to "heat an apple and put it on the counter," then the list of subgoals will include "apple is heated" and "apple on counter." The other two metrics are Path Length Weighted by Success Rate (PLWSR) and Path Length Weighted by Goal Completion (PLWGC), which are SR and GC divided by the length of the episode. For all the metrics, higher is better. We compare our methods, Egocentric Planning Agent (EPA), with other top performers on the current ALFRED leaderboard. Seq2Seq and ET are neural methods that uses end-to-end training (STG[+]20, ZC21). HLSM, FILM, LGS-PRA are hybrid approach that disentangles grounding and modeling (BPF[+]21, MCR[+]22, MC22, IO22).

## 8.1 Results

Table 1 presents a comparison of our outcomes with other top performing techniques on the ALFED dataset. With an unseen test set success rate of 36.07% and a seen test set rate of 44.14%, our EPA method won the last edition of the ALFRED challenge. The ALFRED leaderboard ranks based on SR on unseen data, reflecting the agent's capacity to adapt to unexplored environments. At the moment of writing, our achievements rank us second on the ALFRED leaderboard. The only method surpassing ours is Prompter(IO22), which utilizes the same structure as FILM and incorporates search heuristics-based prompting queries on an extensive language model. However, their own findings indicate that the performance enhancements are largely attributed to increasing obstacle size and reducing the reachable distance represented on the 2D map. The other top performer is LGS-PRA, which is also based on the FILM architecture, that improves performance via techniques on landmark detection method and local pose adjustment. In contrast, EPA employs a semantic graph, a representation distinct from the top-down 2D map applied by FILM, HLSM, LGS-PRA and Prompter. Our findings suggest that the performance boost over FILM and similar methods stems from our iterative planning approach, which facilitates recovery from failure scenarios via flexible subgoal ordering. Techniques outlined in LGS-PRA and Prompter should be compatible with our approach as well. The integration of these techniques with EPA an interesting area to investigate for future works. Our method exhibits lower PLWSR and PLWGC due to our initial 500 exploration steps. These steps are convenient for our planner to amass a diverse of of objects clusters and action angles to determine a proper expandable initial state. Although a pure planning-based exploration would eventually provide us with all the information required to solve the current task, it tends to excessively exploit known object clusters before exploring unknown areas. We only save object information when they are within immediate reach which means the agent is acting on immediate observations. This is done for convenience of converting observation into symbolic state and to reduce the plan length generated by the planner. Additional integration with a top-down SLAM map should help decrease the initial exploration steps, consequently enhancing both PWSR and PLWGC.

| Failure Modes | Seen | Unseen |
|---|---|---|
| Obj not found | 19.36 | 23.54 |
| Collison | 9.14 | 11.34 |
| Interactions fail | 7.33 | 8.98 |
| Obj in closed receptcle | 18.21 | 16.33 |
| Language error | 17.92 | 21.31 |
| Others | 28.04 | 18.5 |

Table 2: Percentage Error of each types in the validation set.

| Ablation | Valid Unseen | | Valid Seen | |
|---|---|---|---|---|
| | SR | GC | SR | GC |
| Base Method | 40.11 | 44.14 | 45.78 | 51.03 |
| w/ gt segment | 45.13 | 49.72 | 49.22 | 53.21 |
| w/ gt depth | 41.85 | 46.03 | 48.98 | 51.46 |
| w/ gt language | 52.29 | 56.39 | 60.55 | 64.76 |
| w/ all gt | 58.33 | 63.71 | 68.49 | 73.35 |

Table 3: Ablation study with ground truth(w/ gt) of different perceptions

| Ablation - parameters | Valid Unseen | | | |
|---|---|---|---|---|
| | SR | GC | PLWSR | PLWGC |
| Base Method | 40.11 | 44.14 | 2.04 | 3.71 |
| w/o init observation | 30.75 | 36.16 | 17.34 | 21.93 |
| 200 init observation | 37.3 | 42.23 | 12.41 | 16.34 |
| w/o object pref | 32.89 | 35.95 | 3.05 | 3.81 |
| w/o fault recovery | 27.31 | 26.55 | 2.41 | 3.36 |
| w/o symbolic planning | 3.47 | 4.61 | 2.16 | 2.98 |

Table 4: Ablation study with different parameters

## 8.2 Ground Truth Ablation

Our model, trained on the ALFRED dataset, incorporates neural perception modules to understand potential bottlenecks in visual perception and language, as explored via ablation studies detailed in Table 3. We observed modest enhancements when providing the model with ground truth depth and segmentation, with ground truth language offering the most significant improvement (18.22% and 19.87%). Our egocentric planner, responsible for generating dynamic policies through observation and interaction, proves more resilient to interaction errors compared to policy-template based methods like HLSM and FILM (MCR$^+$22, BPF$^+$21). Thus, having the correct interpretation of the task description impacts our method more significantly than ground truth perception. We examined common reasons for failed episodes using both the seen and unseen datasets. The most prevalent error (23.54% unseen) is the inability of the agent to locate objects of interest due to failed segmentation or an unreachable angle. Errors stemming from language processing, where the predicted task deviates from the ground truth, are given priority. The results can be seen in Table 2.

## 8.3 Parameter Ablation

Neural vision modules may struggle with out-of-distribution data, such as ALFRED's unseen environments, leading to potential errors in object segmentation and distance estimation. Unlike hybrid methods like FILM and HLSM, our goal-oriented egocentric planner reassesses the environment after failure, generating new action sequences. For instance, in a task to hold a book under a lamp, a failure due to obscured vision can be addressed by altering the action sequence, as illustrated by the drop in performance to 27.31% in our ablation study (Table 4).

FILM's semantic search policy, trained to predict object locations using top-down SLAM maps, differs from our symbolic semantic graph representation, which precludes us from adopting FILM's approach. Instead, we hand-crafted two strategies to search areas around objects and receptacles based on common sense knowledge, which improved our Success Rate by 7.22% with no training. However, our agent only forms a node when it visits a location, necessitating an initial observation phase. Neglecting this step resulted in a 9.35% performance drop. FILM's SLAM-based approach can observe distant objects, requiring fewer initial observation steps and leading to better PLWSR and PLWGC. Combining a top-down SLAM map with our semantic graph could mitigate the need for our initial observation step and improve these scores.

Our algorithm's frequently re-plans according to new observation which is also essential for real-world applications using online planning system. Using modern domain-independent planners, such

as Fast Downward and Fast Forward (Hof01, Hel06), we've significantly reduced planning time. We tested a blind breadth-first search algorithm but found it impractical due to the time taken. A 10-minute cut-off was introduced for each task, which didn't impact our results due to an average execution time of under 2 minutes. Yet, this constraint severely lowered the success rate of the blind search method, only completing 3.47% of the validation unseen tasks (Table 4).

## 8.4 Generalization to Other Tasks

Our approach allows zero-shot generalization to new tasks within the same set of objects and relationships, offering adaptability to new instructions without manually specify new task types and templates. While neural network transfer learning remains challenging in embodied agent tasks, and FILM's template system requires hand-crafted policies for new tasks, our egocentric planning breaks a task into a set of goal conditions, autonomously generating an action sequence. This allows our agent to execute tasks without transfer learning or new task templates. We have chosen environments in which all objects for each task have been successfully manipulated in at least one existing ALFRED task. This selection is made to minimize perception errors. We tested this on five new task types not in the training data, achieving an 82% success rate, as shown in Table 6, sectionC.2 of the Supplementary Material. We could also adapt to new objects, actions, and constraints, such as time and resource restrictions. However, the current ALFRED simulator limits the introduction of new objects and actions. This flexibility to accommodate new goals is a key advantage of our planning-based approach.

## 9 Discussion

Our contribution poses Embodied Agents as acting in a Partially Observable Markov Decision Process (POMDP), which allows us to tackle a broad class of problems, and further expands on the potential applications of planning-based methods for embodied agents. By adopting this approach, our method offers a more flexible and adaptable solution to address the challenges and limitations associated with fixed-policy and handcrafted methods.

We found that using intentional models with a factored representation in embodied agents is crucial. Whereas a learned world model provides a degree of useful information, their associated policies might under-perform for long horizon high-level decision-making, unless they are trained on massive amounts of interactions (SSS[+]17). Comparing a world model with classical planning uncovers interesting distinctions. While creating a PDDL-based solution requires an initial investment of time, that time is amortized in contexts that require quality control per type and skill. On the other hand, the cost spent on crowdsourcing or interacting with a physical environment can be significant (STG[+]20).

Learning classical planning models is relatively straightforward with given labels for objects and actions (AFP[+]18, CDVA[+]22). On the other hand, if we aim to learn a world model, we should consider the performance at planning time and end-to-end solutions may not be the ideal approach in this context. When compared to intentional symbolic-based methods, end-to-end strategies tend to compound errors in a more obscure manner. Furthermore, as learned world models rely on blind search or MCTS, in new tasks the model might waste search time in a subset of actions that are irrelevant if action space is large, while symbolic planning can obtain long plans in a few seconds. We also propose addressing challenging planning tasks using simpler planning techniques, providing a foundation for more complex planning paradigms. Our method could be further enhanced by encoding explicit constraints and resources, such as energy.

While other methods might outperform our model in the ALFRED benchmark, they are often limited in scope. For instance, despite its high unseen success rate, the Prompter model serves as a baseline rather than a comprehensive solution for embodied agents. Our contribution provides a method that can adapt to shifts in task distribution without changing the symbolic planning model, while also supports changes in the agent's capabilities by modifying the symbolic planning model.

We acknowledge certain limitations in our approach. While we support safe exploration assuming reversible actions, real scenarios beyond ALFRED could include irreversible actions and failures. Our model is also sensitive to perception errors, making end-to-end methods potentially more robust in specific scenarios. Furthermore, our perception model lacks memory, potentially hindering nuanced

belief tracking. Despite these challenges, our method offers a promising direction for future research in embodied agent applications. In section B, we expand on the discussion of our approach.

## 10    Conclusion and Future Work

Our work introduces a novel iterative replanning approach that excels in embodied task solving, setting a new baseline for ALFRED. Using off-the-shelf automated planners, we improve task decomposition and fault recovery, offering better adaptability and performance than fixed instruction and end-to-end methods. Moving forward, we aim to refine our location mapping for precise environment understanding, extract more relational data from instructions, and integrate LLMs as exploration heuristics. We plan to tackle NLP ambiguity in multi-goal scenarios by prioritizing and pruning goals based on new information. Moreover, we aim to upgrade our egocentric planner to manage non-deterministic effects, broadening our method's scope beyond ALFRED. This direction, leveraging efficient planning methods for non-deterministic actions, is promising for future research (MBM14).

**Acknowledgments**

We want to acknowledge the ServiceNow research for providing the research opportunity and computational infrastructure to conduct this research project.This work was also made possible due to the assistance of Cyril Ibrahim at ServiceNow during the experimental execution phases of our project. We would also like to thank members of Mulab at Queen's University for providing a supporting research environment. Lastly, we want to acknowledge the assistance from organizers of the CVPR Embodied Agent Workshop for their guidance and feedback throughout the competition(DBB$^+$22).

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

# A Supplementary Material

This Supplementary Material section provides additional details and discussions to complement our main paper. In section B, we expand on the discussion of our approach. Section C presents specific results for the seven tasks of the ALFRED challenge as well as new tasks that can be solved in the environment. Section D provides a formal definition of Object-Oriented Partially Observable Markov Decision Processes (OO-POMDPs), while section E provides a detailed background for our planning approach. Finally, section F provides an in-depth explanation of our method for the ALFRED challenge.

# B Further Discussion

In section 9, we touched on various aspects of our methodology. This section provides more detail on these areas. Our approach to the ALFRED challenge hinges on egocentric planning and a user-defined symbolic planning model. We refer to these models as intentional, as they are purposefully crafted by the user, integrating specific environmental objects and properties. Symbolic models function as a form of world model that facilitates planning. Some methods seek to learn a latent representation, but this often comes with a high sample complexity (HPBL23). Modular meta-learning can reduce the sample complexity (ALPK18, ORCC21, MvSE22). If cost is not a constraint, world models can become components of efficient strategies trained to solve problems in specific domains (SAH$^+$20).

However, with limited resources or data, generalizing to new situations can be challenging. Take an embodied agent in an environment (def D.8) with $N$ skills and $M$ types of objects, tasked with carrying out a sequence of tasks that require plans involving a new combination of $k$ skills and $m$ types of objects, where $n \ll N$ and $m \ll M$[3]. Such setting is within the scope of methods like meta-learning or continual learning for adapting the latent representation for such unfamiliar environments (FAL17). The priors in some modular meta-learning approaches are related with our notions of objects and actions, but their inference algorithms might not scale well for long plans or ignore that some users might also want intentional modelling (ALPK18, ORCC21, MvSE22). Egocentric planning, on the other hand, may provide predictable scalability in these scenarios. While the definition of open-world generalization might be more elusive, our method leans towards systematic generalization across actions and object types.

Planning using certain latent representations or world models can also prove costly. As explained by (AKFM22) (LATPLAN), not all factored representations afford the same level of scalability. The authors of LATPLAN apply variational autoencoders to learn a discrete latent representation, incorporating additional priors so the representation can be interpreted as a STRIPS symbolic model, the most basic type of symbolic model we use for egocentric planning. Symbolic models like STRIPS enable solvers with greater scalability and generalization.

An alternative approach involves methods that learn a symbolic planning model (AFP$^+$18, CDVA$^+$22). Object-oriented POMDPs are especially suitable as the grounding on objects and action labels allows for more efficient algorithms (CMW13, MCRW09), an advantage not enjoyed by the authors of LATPLAN as they only use pairs of images as training data. Learning a symbolic model helps manage the risk of compounding errors since the model can be debugged and improved. Real-world applications may require monitoring the agent's behaviour for specific objects or actions. Furthermore, specific actions may carry inherent risks that need to be avoided. In general, we anticipate that developers of embodied agents will use as much learning as possible, as much intentional modelling as necessary, and symbols to blend the two of them (KSV$^+$22).

Further integration between symbolic planning models and other ML models might help to reduce some limitations of our approach. For instance, in ALFRED, searching for a pan should make the agent head toward the kitchen. If another model like an LLM finds it likely that the pan is in the kitchen, we can use the scores to produce action cost in a similar way to the integration done in SayCan, (iBC$^+$23).

Egocentric planning connects to a body of work aimed at tackling expressive planning problems by harnessing the scalability of classical planners. While egocentric planning focuses on the partial observability of objects, (APG09, PG09) consider partial observability given a set of possible initial

---

[3]While the state space is unknown to the agent, it is aware of the actions and object types.

states and deterministic actions, and (MMB14) considers full observability but with nondeterministic actions. Though these strategies can provide complete solutions proven offline, they can also serve as approximate methods for solving online problems. When planning with probabilistic effects, replanning assuming randomized determinization of actions can be a strong approach if there are no dead ends (YFG07).

Our symbolic planning models are extensions of STRIPS, but other options exist. Hierarchical task networks (HTN) (NAI$^+$03, GA14), for example, offers an alternative to STRIPS as they allow the specification of plans at various abstraction levels. Sometimes, it is more practical to outline potential plans using formalisms like linear temporal logic (LTL) formulae (DV13). Nonetheless, the relationship between these models has been examined, and it is feasible to integrate them with STRIPS-based planning (CC04, HBB$^+$19, MH13).

# C    Results in ALFRED's Task and New Tasks

## C.1    ALFRED Task Performance

| ALFRED Tasks | Valid Unseen | | Valid Seen | |
|---|---|---|---|---|
| | SR | GC | SR | GC |
| look_at_obj_in_light | 53.77 | 54.27 | 47.22 | 30.11 |
| pick_and_place_simple | 43.14 | 47.11 | 37.15 | 39.77 |
| pick_cool_then_place_in_recep | 46.76 | 45.82 | 47.01 | 48.89 |
| pick_clean_then_place_in_recep | 38.77 | 41.37 | 38.01 | 42.21 |
| pick_cool_then_place_in_recep | 46.26 | 46.81 | 47.21 | 47.33 |
| pick_heat_then_place_in_recep | 37.69 | 40.50 | 39.82 | 41.33 |
| pick_two_obj_and_place | 30.33 | 35.71 | 27.37 | 31.21 |

Table 5: Performance on Valid Unseen and Valid Seen for each task

To analyze the strengths and weaknesses of our approach in different types of tasks, we conducted further evaluations by task type as in Table 5. Our findings reveal several noteworthy observations. Firstly, the success rates (SR) and goal completion rates (GC) for the "Pick_two_obj_and_place" task are lower than the rest, mostly due to these tasks involving more objects and with smaller masks. The task "look_at_obj_in_light" has the best performance due to the variety of lamp is low compare to other common objects. Secondly, there is no strong correlation between the number of interactions involved in a task type and its performance (STG$^+$20). For example, the task "pick_cool_then_place_in_receptacle" has an extra cooling step compare to "pick_and_place_simple" but still have better performance. A detailed description of each task can be found in the original ALFRED paper on page 16 to 19 (STG$^+$20).

## C.2    New Task Performance

In section 8.4, we discuss the generalization of our approach to new tasks. The results are summarized in table 6.

| | SR | GC | PLWSR | PLWGC |
|---|---|---|---|---|
| Pickup and Place Two Objects | 90.00 | 93.44 | 3.71 | 4.37 |
| Clean-and-Heat Objects | 80.00 | 89.77 | 2.56 | 3.90 |
| Clean-and-Cool Objects | 85.00 | 93.38 | 4.44 | 4.78 |
| Heat-and-Cool Objects | 80.00 | 89.20 | 4.01 | 4.66 |
| Pick and Place Object in Drawer | 70.00 | 77.81 | 2.59 | 3.22 |

Table 6: New tasks

Our approach enable us to reuse the definition of existing models to achieve zero-shot generalization to new tasks as long as the set of actions, objects and their relationships remain the same. In many embodied agent applications, agent need to have ability to adapt to new tasks. For instance, organizing a living room could involve storing loose items on a shelf or arranging them on a table. An

agent familiar with common household objects and equipped with the skills to handle them should seamlessly manage both types of cleaning tasks. However, transfer learning for neural networks remains challenging in the context of embodied agent tasks generalizing across objects and skills. Similarly, the template system used in FILM demands hand-crafted policies for new task types. The ALFRED dataset include instructions for each tasks that (STG$^+$20), although our approach does not use them. In a our egocentric planning formulation, a task can be broken down into a set of goal conditions that comprise objects of interest. The planner can then autonomously generate an action sequence to reach these goals. Consequently, our method can independently execute these tasks without the need for transfer learning or the addition of new task templates. To demonstrate this, we devised 5 new task types that were not present in the training data. We have chosen environments in which all objects for each task have been successfully manipulated in at least one existing ALFRED task. This selection is made to minimize perception errors. These tasks share the same world model as ALFRED and thus do not necessitate any additional modification beyond specifying a new goal. We selected environments and layouts from the training set where subgoals required for each new task type are feasible, ensuring the feasibility of the new task for our agent. Table 6 lists these new tasks and their success rates. We selected 20 scenarios for each new task and maintained the maximum 10 interaction error as in the ALFRED problem. Our method was able to achieve an impressive overall success rate of 82% without any additional adjustments beyond the new goal specification. However, due to the constraints of the ALFRED simulator, we're unable to experiment with introducing new objects and action types beyond what's available. For instance, "heating" in ALFRED is solely accomplished via a microwave. In real-world scenarios, we could easily incorporate ovens and actions to operate them with minor alterations to our PDDL domains. Similarly, we could introduce time and resource restrictions for each action, which are common in many PDDL benchmarks. The ability to accommodate new goals and constraints is a key advantage of our planning-based approach.

### C.3 Computation

Perception and Language module was fine-tuned on a Nvidia 3080. The planning algorithm uses a 8 core Intel i7 CPU with 32 GB of ram. More detailed on traning can be found in (MCR$^+$22)

## D  Embodied Agents Generalizing within given Objects Types

In this section, we provide further details about embodied agents presented in section 4. Embodied agents perceive and act in environments to accomplish tasks. Navigating challenges and constraints imposed by their environment and tasks, these agents ensure safety and adhere to physical restrictions. Taskable agents rely on their surroundings and inherent capabilities, creating a strong dependency and often specializing in particular environments. This specialization shapes their cognition and problem-solving strategies, presenting an opportunity for robust agent design.

We exploit this limitation by focusing on generalization across tasks involving the same object types and their relationships, applicable to both physical and software agents. Taskable agents possess specialized capabilities for these object types, as seen in robotics where battery level, body state, and actuators are distinct variables (CST$^+$22). Object types also serve as inputs/outputs in end-to-end approaches with learned representations and appear in embodied agents operating in software environments, using APIs and maintaining internal states. Our symbolic planning approach aims to provide a basis for designing adaptable and flexible agents that sense and act upon given object types. We start by defining object types, and entities grounded in these types.

**Definition D.1** (Objects and Object Types). An *Object Type*, denoted as $\langle \mathcal{T}, \mathcal{V} \rangle$, consist of a finite set of object types $\mathcal{T}$, along with a mapping $\mathcal{V} : t_i \to \mathcal{V}_i$ that assigns each type $t_i \in \mathcal{T}$ a set of possible values $\mathcal{V}_i$. An *Object* $o = (t_i, v)$ is an *instance* of a type with a specific value, i.e., $t_i \in \mathcal{T}$ and $v \in \mathcal{V}(t_i)$.

**Example D.2** (Object types and Values). *An example of an object type is $t_{objectSeen}$ with values $\mathcal{V}_{(x,y)} \times \mathcal{V}_{direction} \times \mathcal{V}_{objectSubtype} \times \mathcal{V}_{id}$ where $\mathcal{V}_{(x,y)}$ is position in a grid, $\mathcal{V}_{direction}$ is the direction the object is facing, $\mathcal{V}_{objectSubtype}$ is the category of the object observed, and $\mathcal{V}_{id}$ is a unique identifier for the object. Other related examples of object types are $t_{movable}$ with values $\mathcal{V}_{movableSubtype} \times \mathcal{V}_{id} \times \mathcal{V}_{objectproperties}$, and $t_{obstacle}$ with values $\mathcal{V}_{obstacleSubtype} \times \mathcal{V}_{id} \times \mathcal{V}_{objectproperties}$. Furthermore, an action* `pickup` *with parameters $\Phi_{\mathcal{E}}(a) = (t_{objectSeen})$ is an action that picks up a specific object.*

Object types are used as parameters of entities of actions of Object-oriented POMDP defined, and as parameters of actions and predicates in planning domains in section E.

**Definition D.3** (Typed Entity)**.** Given a set of entities $\Xi$, object types $\langle \mathcal{T}, \mathcal{V} \rangle$, and a mapping $\Phi : \xi \to \bigcup_{k \in \mathbb{N}} \mathcal{T}^k$. A *Type Entity* $\xi \in \Xi$ is mapped to $\Phi(\xi) = (t_1, t_2, ..., t_{k_\xi})$, with each $t_i \in \mathcal{T}$. This mapping associates each entity with a $k_\xi$-tuple of object types, corresponding to $k_\xi$ parameters of the entity, defining the *arity* of the entity $\xi$. We refer to the variables $var \in \{var_1, var_2, ..., var_{k_\xi}\}$ as the positional parameters of the entity $k_\xi$, denoted $\xi(var_1, var_2, ..., var_k)$.

**Definition D.4** (Grounded Entity)**.** Given a set of typed entities $\Xi$, with object types $\langle \mathcal{T}, \mathcal{V} \rangle$, and a mapping $\Phi$, a *Grounded Entity*, denoted as $\xi^g$ for $\xi \in \Xi$, is a parametric object where each type argument of $\xi$ specified by $\Phi(\xi)$ is replaced by an object of the corresponding type. Specifically, $\xi^g = \xi(o_1, o_2, ..., o_{k_\xi})$, where each $o_i = (t_i, v_i)$ with $v_i \in \mathcal{V}(t_i)$. The set of all grounded entities is denoted by $\Xi^g$.

Now we define the environment where the embodied agent operates as a Partially Observable Markov Decision Process (POMDP) with costs. Cost POMDPs are related to classical MDPs but feature partial observability and use costs instead of rewards (MK12, GB13).

The literature sometimes confound the environment with the underlying the POMDP, but we want to make explicit what is available to the agent. Thus, while our underlying model is a POMDP, we consider agents that can aim to generalize without being fully aware of the distribution of future tasks, except for the actions and object types that are explicit in the environment. Therefore, we abuse slightly the notation to annotate with the objects defined below with the environment, denoted $\mathcal{E}$.

**Definition D.5** (Cost POMDP)**.** A *Cost POMDP* is a tuple $\langle S_\mathcal{E}, A_\mathcal{E}, T_\mathcal{E}, C_\mathcal{E}, P_\mathcal{E}, Z_\mathcal{E} \rangle$, where $S_\mathcal{E}$ is a set of states, $A_\mathcal{E}$ is a finite set of actions, $T_\mathcal{E} : S_\mathcal{E} \times A_\mathcal{E} \times S_\mathcal{E} \to [0, 1]$ is the transition function, $C_\mathcal{E} : S_\mathcal{E} \times A_\mathcal{E} \to \mathbb{R} > 0$ is the cost function, $P_\mathcal{E}$ is a set of perceptions, and $Z_\mathcal{E} : S_\mathcal{E} \times A_\mathcal{E} \times P_\mathcal{E} \times S_\mathcal{E} \to [0, 1]$ is the sensor function.

Object-oriented POMDP is an extension of a Cost POMDP that includes objects and their properties to be used for expressing perceptions and actions.

**Definition D.6** (Object-oriented POMDP)**.** An *Object-oriented POMDP* $\Gamma$, represented by $\langle S_\mathcal{E}, A_\mathcal{E}, \Phi_\mathcal{E}, T_\mathcal{E}, C_\mathcal{E}, P_\mathcal{E}, Z_\mathcal{E}, \mathcal{T}_\mathcal{E}, \mathcal{V}_\mathcal{E} \rangle$, extends a Cost POMDP by incorporating a predefined set of *Object Types* $\langle \mathcal{T}_\mathcal{E}, \mathcal{V}_\mathcal{E} \rangle$. It defines the set of perceptions $P_\mathcal{E}$ where a perception $p \in P_\mathcal{E}$ is a finite set of objects $\{o_1, o_2, \ldots, o_m\}$, and the set of typed actions $A_\mathcal{E}$ defined for the object types $\langle \mathcal{T}_\mathcal{E}, \mathcal{V}_\mathcal{E} \rangle$ and the map $\Phi_\mathcal{E}$.

In this context, *Grounded Actions* refer to grounded entities, where the set of entities is the actions $A_\mathcal{E}$. Values in object-oriented POMDPs can be dense signals like images or sparse signals like object properties. Values can have any structure, so they can refer to values in the domain of other types. Indeed, Object-oriented POMDPs can express Relational POMDPs (WK10, WJK08, SB09, TGD04, WPY05, DNR⁺20). However, our approach emphasizes the compositionality of the objects and their properties.

Agents interact with the POMDP through an environment. Given a task, we initialize the environment using the task details. The agent receives the tasks and acts until it achieves the goal or fails due to some constraint on the environment. The agent achieves a goal if, after a sequence of actions, the environment's hidden state satisfies the goal.

**Definition D.7** (Task)**.** Given an Object-oriented POMDP $\Gamma$, a *Task* is a tuple $T_\mathcal{E} = \langle I_\mathcal{E}, G_\mathcal{E} \rangle$, where $I_\mathcal{E}$ is an object that the environment interpret for setting the initial state, and $G_\mathcal{E}$ is a set of goal conditions expressed as objects $\{g_1, g_2, \ldots, g_n\}$, with each $g_i$ being an instance of an object.

**Definition D.8** (Object-oriented Environment)**.** Given a hidden Object-oriented POMDP $\Gamma$, an *Object-oriented Environment*, or just *Environment*, is a tuple $\mathcal{E} = \langle A_\mathcal{E}, \mathcal{T}_\mathcal{E}, \mathcal{V}_\mathcal{E}, \text{reset}, \text{step} \rangle$, where $A_\mathcal{E}, \mathcal{T}_\mathcal{E}$, and $\mathcal{V}_\mathcal{E}$ are as previously defined, $\text{reset} : T_\mathcal{E} \to P_\mathcal{E}$ is a function that takes a task $T_\mathcal{E} = \langle I_\mathcal{E}, G_\mathcal{E} \rangle$, resets the environment's internal state using $I_\mathcal{E}$ and returns an initial observation, and $\text{step} : A_\mathcal{E}^g \to P_\mathcal{E} \times \mathbb{R}$ is the interaction function that updates the environment's internal state according to the execution of a grounded action $a_g$, for $a \in A_\mathcal{E}$, and returns an observation and a cost.

In some environments, $I_\mathcal{E}$ might be opaque for the agent. In others, it might containing information not used by the environment like user's preferences about how to complete a task. However, we abuse this notion and refer to $I_\mathcal{E}$ as the initial state of the environment.

Separating Object-oriented POMDPs from environments allows us to consider the scope of the generalization of our agents within the same POMDP. While the agent cannot see the state space directly, it is aware of the actions and the object types.

**Definition D.9** (Embodied Agent). Given an environment $\mathcal{E} = \langle A_\mathcal{E}, \mathcal{T}_\mathcal{E}, \mathcal{V}_\mathcal{E}, \mathsf{reset}, \mathsf{step} \rangle$, and a task $T_\mathcal{E} = \langle I_\mathcal{E}, G_\mathcal{E} \rangle$, an *Embodied Agent* is a tuple $\langle M_I, \pi_M, M_U, G_M \rangle$, where $M_I : P_\mathcal{E} \times I_\mathcal{E} \times G_\mathcal{E} \to \mathcal{M}$ is a function that takes an initial observation, a task, and returns the initial internal state of the agent, $\pi_M : \mathcal{M} \to A_\mathcal{E}^g$ is the agent's policy function that maps mental states to a grounded action in $A_\mathcal{E}^g$, $M_U : \mathcal{M} \times A_\mathcal{E}^g \times P_\mathcal{E} \times \mathbb{R} \to \mathcal{M}$ is the mental state update function receiving a grounded executed action, the resulting observation, and the cost, and $G_M : \mathcal{M} \to [0, 1]$ is the goal achievement belief function that returns the belief that the agent has achieved the goal based on its mental state $M$.

Algorithm 2 provides a general framework for an agent's execution process. By instantiating the policy, mental state update function, and goal achievement belief function using appropriate techniques, the agent can adapt to a wide range of environments and tasks. Although new tasks can set the agent in unseen regions of the state space $S_\mathcal{E}$, the agent remains grounded on the known types and the actions parameterized on object types, $\mathcal{T}_\mathcal{E}$. They offer opportunities for generalization by narrowing down the space of possible policies and enabling compositionality. The agent's mental state $M$ can include its belief state, internal knowledge, or other structures used to make decisions. The policy $\pi_M$ can be a deterministic or stochastic function, learned from data or user-specified, or a combination of both. As the agent has an internal state, its policy can rely on a compact state or on the history of interactions.

---

**Algorithm 2** Agent Execution

**Input:** Environment $\langle A_\mathcal{E}, \mathcal{T}_\mathcal{E}, \mathcal{V}_\mathcal{E}, \mathsf{reset}, \mathsf{step} \rangle$
**Input:** Agent $\langle M_I, \pi_M, M_U, G_M \rangle$
**Input:** Task $\langle I_\mathcal{E}, G_\mathcal{E} \rangle$
**Input:** Probability threshold $\theta$
**Output:** Successful trace $\tau$ or Failure
1:   $p \leftarrow \mathsf{reset}(I_\mathcal{E}, G_\mathcal{E})$
2:   $\tau = []$            $\triangleright$ Empty trace
3:   $M \leftarrow M_I(p, I_\mathcal{E}, G_\mathcal{E})$
4:   **while** $G_M(M) < \theta$ **do**
5:      $a^g = a(o_1, o_2, ..., o_{k_a}) \sim \pi_M(M, A_\mathcal{E}^g)$
         $\triangleright$ where $a$ in $A_\mathcal{E}$ and $o_i$ are objects.
6:      $\tau.\mathsf{append}(a^g)$
7:      $p', c \leftarrow \mathsf{step}(a^g)$
8:      **if** $failed \in p'$ **then**
9:          **return** Failure
10:     $M \leftarrow M_U(M, a^g, p', c)$
11: **return** $\tau$

---

The agent starts in the initial state set from $I_\mathcal{E}$ and aims to achieve the goal $G_\mathcal{E}$. While the agent does not know the true state of the environment, agents can aim to generalize across actions and the object types as they are always available. While we prefer policies with lower expected cost, our main concern is achieving the goal. It is possible that a task might be unsolvable, possibly because of actions taken by the agent. We assume that failures can be detected, for instance, by receiving a special observation type $failed$. We further assume that after failure, costs are infinite, and $failed$ is always observed.

The Iterative Exploration Replanning (IER), Algorithm 1 in section 5, is a instance of algorithm 2. Instead of using a parameter $\theta$, it considers the problem solved when it obtains a plan that achieves the goal.

## E    Detailed Background: Parameterized Full-Observable Symbolic Planning

In this section, we describe the background definitions of parameterized full-observable symbolic models mentioned in section 5.1. We define planning domains and problems assuming full observability, including classical planning that further assumes deterministic actions, also known as STRIPS (GNT04, GB13). Planning domains and problems rely on typed objects (Def D.1), as object-oriented POMDPs, but planning actions include a model of the preconditions and effects.

**Definition E.1** (Parametric Full-Observable Planning Domain). A *parametric full-observable planning domain* is a tuple $\mathcal{PD} = \langle \mathcal{T}, \mathcal{V}, \mathcal{P}, \mathcal{A}, \Phi \rangle$, where $\langle \mathcal{T}, \mathcal{V} \rangle$ is a set of object types, $\mathcal{P}$ is a set of typed predicates and $\mathcal{A}$ is a set of typed actions, both typed by the object types and the map $\Phi$. Each action $a$ has an associated $\mathrm{PRE}_a$, expressing the preconditions of the action, and $\mathrm{EFF}_a$, expressing its effects. For each grounded action $a(o_1, o_2, ..., o_k) \in A^g$, the applicability of the grounded action in a state $s$ is determined by checking whether the precondition $\mathrm{PRE}_a$ is satisfied in $s$. The resulting state after applying the grounded action is obtained by updating $s$ according to the effect update $\mathrm{EFF}_a$. The specification of $\mathrm{EFF}_a$ is described below.

While the values in both parametric full-observable planning domains and object-oriented POMDPs can be infinite (def D.6), the values used in concrete planning problems are assumed to be finite. As a consequence, the set of grounded actions of a planning problems is finite (def D.4).

**Definition E.2** (Parametric Full-Observable Planning Problem). A *parametric full-observable planning problem* is a tuple $\langle \mathcal{PD}, \mathcal{O}, I, G \rangle$, where $\mathcal{PD} = \langle \mathcal{T}, \mathcal{V}, \mathcal{P}, \mathcal{A}, \Phi \rangle$ is a parametric full-observable planning domain, $\mathcal{O}$ is a finite set of objects, each associated with a type in $\mathcal{T}$, $I$ is a description of the initial state specifying the truth value of grounded predicates in $\mathcal{P}^g$, $G$ is a description of the goal specifying the desired truth values of grounded predicates in $\mathcal{P}^g$, where $\mathcal{P}^g$ are the predicates $\mathcal{P}$ grounded on object types $\langle \mathcal{T}, \mathcal{V} \rangle$ and the map $\Phi$ encoding their typed arguments.

This definition covers a wide range of symbolic planning problems featuring full observability including probabilistic effects (KHW95). In particular, the definition can be specialized for classical planning problems, where the actions are deterministic, corresponding to STRIPS, to the most studied symbolic model (GB13).

For classical planning, given an action $a \in \mathcal{A}$ parameterized by a list of object types $a(t_1, t_2, ..., t_k)$, their precondition or effect are formulas $\gamma$ expresse in terms of the variables $var \in \{var_1, var_2, ..., var_k\}$ that refer to positional parameters of the action. We denote the grounded formula $\gamma(o_1, o_2, \ldots, o_k)$ as $\gamma$ with each variable $var_i$ that occurs in $\gamma$ is replaced by corresponding objects $o_i$.

**Definition E.3** (Parametric Classical Planning Domain). A *parametric classical planning domain* is a parametric full-observable planning domain where each action $a \in \mathcal{A}$ has preconditions $\text{PRE}_a$, and an effect update $\text{EFF}_a$ represented as a tuple $(\text{ADD}_a, \text{DEL}_a)$ of add and delete effects, all sets of predicates in $\mathcal{P}$. Given a grounded action $a(o_{arg})$ with $o_{arg} = o_1, o_2, \ldots, o_k$, the precondition $\text{PRE}_a$ is satisfied in a state $s$ if $\text{PRE}_a(o_{arg}) \subseteq s$. The effect update $\text{EFF}_a$ describes the resulting state as $(s \setminus \text{DEL}_a(o_{arg})) \cup \text{ADD}_a(o_{arg})$ after applying the action.

**Definition E.4** (Parametric Classical Planning Problem). Given a parametric classical planning domain $\mathcal{PD}$, a *parametric classical planning problem* is a parametric full-observable planning problem where $I$ and $G$ are conjunctions of predicates in $\mathcal{P}$ (equivalently viewed as sets).

As Egocentric Planning relies on replanning over a sequence of related planning problems, it is convenient to define some operations required for exploration, as the agent discovers more objects. They allow us to implement the function *ExploreAct* in Algorithm 1.

**Definition E.5** (Precondition Extension). Given a parametric full-observable planning problem and an action $a \in \mathcal{A}$, *extending the precondition* $\text{PRE}_a$ with $\Delta\text{PRE}_a$, denoted $\text{PRE}_a \oplus \Delta\text{PRE}_a$, is a new precondition that is satisfied in a state $s$ if $\text{PRE}_a$ and $\Delta\text{PRE}_a$ are both satisfied in $s$.

**Definition E.6** (Effect Extension). Given a parametric full-observable planning problem and an action $a \in \mathcal{A}$, *extending the effect* $\text{EFF}_a$ with $(\Delta\text{ADD}_a, \Delta\text{DEL}_a)$, denoted $\text{EFF}_a \oplus (\Delta\text{ADD}_a, \Delta\text{DEL}_a)$, is a new effect that applied to a state $s$ results the new state $s'$ that is like $s$ but with modifications $\text{EFF}_a$, then $\Delta\text{DEL}_a$ is not true in $s'$, and $\Delta\text{ADD}_a$ is true in $s'$, in that order, assuming that neither $\Delta\text{DEL}_a$ nor $\Delta\text{ADD}_a$ appear in $\text{EFF}_a$.

With these operations we can define the Egocentric Planning algorithm for parametric full-observable planning problems. For instance, in the case of probabilistic planning the preconditions can be defined as in classical planning, so the precondition update is the same. The effect update of probabilistic actions is a probability distribution over the possible effects. However, extending the effect with a new effect is straightforward as the new effect must enforce that $Pr(\text{ADD}_a|s) = 1$ and $Pr(\text{DEL}_a|s) = 0$.

For classical planning, the precondition extension is the union of the original precondition and the new precondition as both are a set of predicates. The $\text{EFF}_a$ of classical planning actions $(\text{ADD}_a, \text{DEL}_a)$ is a tuple of sets of predicates, so updating the effect amounts to updating both sets of predicates.

**Definition E.7** (Precondition Extension Classical). For parametric classical planning problems, *extending the precondition* $\text{PRE}_a \oplus \Delta\text{PRE}_a$ is $\text{PRE}_a \cup \Delta\text{PRE}_a$.

**Definition E.8** (Effect Update Classical). For parametric classical planning problems, an *update to the effect* $(\text{ADD}_a, \text{DEL}_a)$ with $(\Delta\text{ADD}_a, \Delta\text{DEL}_a)$ is $(\text{ADD}_a \cup \Delta\text{ADD}_a, \text{DEL}_a \cup \Delta\text{DEL}_a)$.

For ALFRED, as the planning domain in classical, *ExploreAct* $(a, o)$ creates a copy of $a$ extended with a new precondition (`unknown` $o$) so $a$ can only be applied if $o$ is still unknown. The effects are extended by deleting (`unknown` $o$) and adding (`explored`), so applying the exploration action satisfies the goal (`explored`).

# F  Approach: Additional Details

In this section, we provide additional details on our approach discussed in section 6. Both the environment and the symbolic planning model consider objects and parametric actions, but they are represented differently.

For instance, our approach involves a vision model that perceives in a scene objects of known classes, such as `apple` or `fridge`, including their depths and locations (see sections 4 and D). Similarly, the planning domain includes corresponding object types, such as `AppleType` and `FridgeType` (see sections 5.1 and E). When the agent perceives an apple at location `l1`, the algorithm creates a fresh object `o1` for the apple, sets its type to `(objectType o1 AppleType)`, and its location to `(objectAtLocation o1 l1)`. Additionally, the domain contains information about what can be done with the apple, such as `(canCool FridgeType AppleType)`. Likewise, while the ALFRED environment contains actions like `toggle`, the planning domain contains actions like `coolObject`, which is parameterized on arguments such as the location, the concrete object to be cooled, and the concrete fridge object. Finally, if the task provided in natural language requests to cool down an apple, the planning problem goal contains, for instance, `(exists (isCooled ?o))`, which is satisfied when some object has been cooled down.

The remainder of this section is structured as follows. First, we provide additional details on the Language and Vision modules, including the specific classes they detect and how they are utilized. Next, we elaborate on the semantic spatial graph and its role in generating the agent's mental model, which tracks object locations as the agent perceives and modifies the environment. We then provide specific details on the planning domain used for ALFRED, which completes the elements used in our architecture depicted in Figure 1. Finally, we conclude the section with additional details on the egocentric planning agent that won the ALFRED challenge.

## F.1  Language and Vision Module

Table 7: List of Objects

| | | | | |
|---|---|---|---|---|
| alarmclock | apple | armchair | baseballbat | basketball |
| bathtub | bathtubbasin | bed | blinds | book |
| boots | bowl | box | bread | butterknife |
| cabinet | candle | cart | cd | cellphone |
| chair | cloth | coffeemachine | countertop | creditcard |
| cup | curtains | desk | desklamp | dishsponge |
| drawer | dresser | egg | floorlamp | footstool |
| fork | fridge | garbagecan | glassbottle | handtowel |
| handtowelholder | houseplant | kettle | keychain | knife |
| ladle | laptop | laundryhamper | laundryhamperlid | lettuce |
| lightswitch | microwave | mirror | mug | newspaper |
| ottoman | painting | pan | papertowel | papertowelroll |
| pen | pencil | peppershaker | pillow | plate |
| plunger | poster | pot | potato | remotecontrol |
| safe | saltshaker | scrubbrush | shelf | showerdoor |
| showerglass | sink | sinkbasin | soapbar | soapbottle |
| sofa | spatula | spoon | spraybottle | statue |
| stoveburner | stoveknob | diningtable | coffeetable | sidetableteddybear |
| television | tennisracket | tissuebox | toaster | toilet |
| toiletpaper | toiletpaperhanger | toiletpaperroll | tomato | towel |
| towelholder | tvstand | vase | watch | wateringcan |
| window | winebottle | | | |

### F.1.1  Language Module

We used a BERT-based language models to convert a structured sentence into goal conditions for PDDL (DCLT19). Since the task types and object types of defined in the ALFRED metadata, we use a model to classify the type of task given a high-level language task description listed in Table 9

Table 8: List of Receptacles

| | | | |
|---|---|---|---|
| Bathtub | Bowl | Cup | Drawer |
| Mug | Plate | Shelf | Sink |
| Box | Cabinet | CoffeeMachine | CounterTop |
| Fridge | GarbageCan | HandTowelHolder | Microwave |
| PaintingHanger | Pan | Pot | StoveBurner |
| DiningTable | CoffeeTable | SideTable | ToiletPaperHanger |
| TowelHolder | Safe | BathtubBasin | ArmChair |
| Toilet | Sofa | Ottoman | Dresser |
| LaundryHamper | Desk | Bed | Cart |
| TVStand | Toaster | | |

Table 9: List of Goals

| | |
|---|---|
| pick_and_place_simple | pick_two_obj_and_place |
| look_at_obj_in_light | pick_clean_then_place_in_recep |
| pick_heat_then_place_in_recep | pick_cool_then_place_in_recep |
| pick_and_place_with_movable_recep | |

and a separate model for deciding objects and their state based on the output of the second model in Table 7 and Table 8. For example, given the task description of "put an apple on the kitchen table," the first model will tell the agent it needs to produce a "pick-up-and-place" task. The second model will then predict the objects and receptacle we need to manipulate. Then the goal condition will be extracted by Tarski and turn in to a PDDL goal like *(on apple table)* using the original task description and output of the both models. We use the pre-trained model provided by FILM and convert the template-based result into PDDL goal conditions suitable for our planner. Although ALFRED provides step-by-step instructions, we only use the top-level task description since they are sufficient to predicate our goal condition. Also, our agent needs to act based on the policy planner, which makes step-by-step instructions unnecessary. This adjustment could also make the agent more autonomous since step-by-step instructions are often not available in many embodied agent settings.

### F.1.2  Vision Module

The vision module is used to ground relevant object information into symbols that can be processed by our planning module. At each time step, the module receives an egocentric image of the environment. The image is then processed into a depth map using a U-Net, and object maks using Mask-RCNN. (HGDG17, LCQ$^{+}$18). Both masks are WxH matrices where W, and N are the width and height of the input image. Each point in the depth mask represents the predicted pixel distance between the agent and the scene. The average depth of each object is calculated using the average of the sum of the point-wise product of the object and the depth mask. Only objects with an average reachable distance smaller than 1.5 meter is stored. The confidence score of which our agent can act on an object is calculated using the sum of their mask. This score gives our egocentric planner a way to prioritize which object to interact with.

### F.2  Semantic Spatial Graph

The spatial graph plays the role of an agent's memory during exploration and bridges the gap between grounded objects and the planner's state representation requirements. The exploration employs an egocentric agent who only has direct access to its local space where action can be executed within a distance threshold of 1.5 meters. The key for each node is encoded as location, with visual observations being the values. The edges represent the agent's actions, comprising four specific actions: MOVEAHEAD, MOVEBACK, TURNLEFT, and TURNRIGHT, with movement actions having a step size of 0.25 meter and turning actions featuring a turn angle of 90°. This spatial graph is updated and expanded over time through policies generated by our egocentric planner, with each node storing object classes, segmentation masks, and depth information produced via the vision module. Fig. 2 shows an example of a semantic spatial graph.

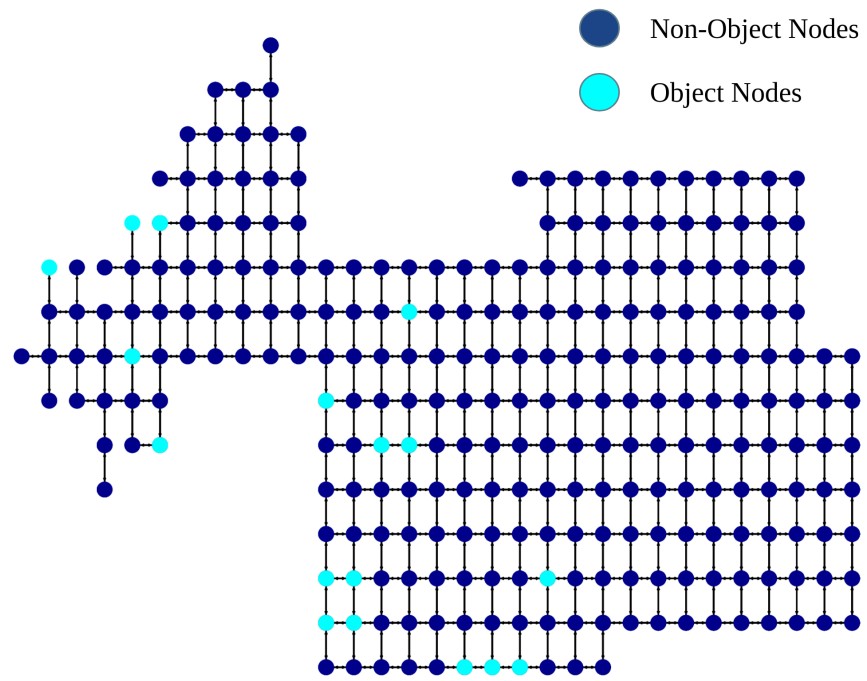

Figure 2: Example of a Semantic Spatial Graph after partial Exploration. The agent is at one of the nodes.

The spatial graph undergoes continuous updates and expansions as a result of the policies generated by our egocentric planner. It involves the extraction and storage of objects within the graph, subject to a confidence score threshold of 0.8. Objects that meet or exceed this confidence score are deemed reliable and considered for storage. Each node within the graph maintains a record of various attributes associated with the objects, including object classes, segmentation masks, and depth information. The object classes provide information about the category or type of each object, allowing for efficient categorization and analysis. The segmentation masks of the extracted objects are stored as a list of pixel coordinates. These masks outline the boundaries of each object within the captured visual data. By storing the masks in this format, we can later retrieve and utilize the spatial information of each object during subsequent analyses or interactions. Moreover, the distance to each object is calculated by leveraging the predicted distance of the top 100 pixels with reference to the depth mask generated by the U-net model. The U-net model is responsible for producing a depth mask that estimates the distances of various points within the scene. By employing the predicted distance values of the top 100 pixels associated with each object, we obtain an approximate measure of its distance from the egocentric viewpoint. Actionable objects are treated as affordances at each node, allowing the agent to perform desired actions. Nodes with objects of interest are ranked based on the confidence scores outlined earlier output by the MaskRCNN.

### F.3 Egocentric Algorithm for ALFRED

The egocentric planning algorithm for ALFRED uses location as the only anchor types $\mathcal{T}_a$, and the move action as the only exploration action $\mathcal{X}$. Consequently, exploration plans employ move actions to visit unexplored locations, revealing new objects visible from such locations. As the agent explores, the mental model creates new objects and adds them to the initial state, facilitating the incremental construction of a semantic spatial graph, which maps the environment and object locations. An ALFRED task, $T_{\mathcal{E}} = \langle I_{\mathcal{E}}, G_{\mathcal{E}} \rangle$, is characterized by a goal that corresponds to one of the seven tasks detailed in section 2, and involves specific predicates for the task-related objects. The goal is expressed as an existential formula over potential objects, as it refers to objects not visible from the initial state (Listings 2).

We also implemented some ALFRED-specific optimizations. For example, we mitigate the risk of executing irreversible actions by only attempting them after devising a plan projected to achieve the goal. This strategy implies that some actions are unavailable during the exploration phase. For each time step, our egocentric planner will produce a goal or exploration action depending on whether the agent has sufficient information to achieve its goal. The replanning loop repeats when an action fails. Priority are given to unknown location based on objects of interest for a particular task and number of total objects found in an area. They are ranked base on number of such objects. Information are gathered for each action taken which is stored in the semantic map. In order to reduce the amount of overall exploration required at the online planning stage, we allow our agent to first conduct 500 random exploration movements. The semantic graph after exploration is used as the initial state for our egocentric planning setup. To enhance efficiency, we tailor the planning domain to the specific task, disregarding actions unlikely to contribute to reasonable plans to achieve the goal. This optimization can be automatically implemented via reachability analysis, a technique commonly used in classical planning to prevent grounding actions irrelevant to achieving the goal from the initial state (Hof01).

## F.4  PDDL Example

Below we provide examples assuming that the natural language description of the task is "heat potato in the microwave and put the potato on the table." We start by generating the planning setup for the egocentric planner, given the task identified as "pick-heat-then-place-in-recep." The objects involved in this manipulation are classified as "MicrowaveType", "PotatoType" and "TableType". The egocentric planner will eventually produce a sequence of actions to accomplish the desired task. In this case, the steps might include picking up the potato, heating it, and then placing it on the table.

Listings 1 and 2 show the PDDL domain and problem files for the example task. In a PDDL domain for classical planning, both predicates and actions arguments are expressed as a list of arguments, each one a variable with an optional type. For instance, (conn ?x1 - location ?x2 - location ?x3 - movement) express a predicate conn with three arguments: two locations and a kind movement: ahead, rotate-left or rotate-right. In a PDDL problem for classical planning, object constants are associated with an optional type, and the initial state is a list of predicates grounded on existing constants. For instance, l1 l2 l3 - location means three constants of type location. The initial state could be (and (at l1) (conn l1 l2 ahead)). An action (MoveAgent ?from ?to - location ?dir - movement) has as a precondition that the agent is at the location ?from and both locations are contiguous in the direction ?dir. Executing the action sets to false that the agent is at the location ?from, and sets to true that the agent is at the location ?to. Classical plans in PDDL are sequences of actions grounded on constants. For instance, a first possible action might be (MoveAgent l1 l2 ahead), leading to the state (and (at l2) (conn l1 l2 ahead)). For an introduction to PDDL see (HLMM19).

Listing 1: ALFRED PDDL domain file

```
1  (define (domain alfred_task)
2      (:requirements :equality :typing)
3      (:types
4          agent - object
5          location - object
6          receptacle - object
7          obj - object
8          itemtype - object
9          movement - object
10     )
11
12     (:predicates
13         (conn ?x1 - location ?x2 - location ?x3 - movement)
14         (move ?x1 - movement)
15         (atLocation ?x1 - agent ?x2 - location)
16         (receptacleAtLocation ?x1 - receptacle ?x2 - location)
17         (objectAtLocation ?x1 - obj ?x2 - location)
18         (canContain ?x1 - itemtype ?x2 - itemtype)
19         (inReceptacle ?x1 - obj ?x2 - receptacle)
20         (recepInReceptacle ?x1 - receptacle ?x2 - receptacle)
```

```
21          (isInReceptacle ?x1 - obj)
22          (openable ?x1 - itemtype)
23          (canHeat ?x1 - itemtype ?x2 - itemtype)
24          (isHeated ?x1 - obj)
25          (canCool ?x1 - itemtype ?x2 - itemtype)
26          (isCooled ?x1 - obj)
27          (canClean ?x1 - itemtype ?x2 - itemtype)
28          (isCleaned ?x1 - obj)
29          (isMovableReceptacle ?x1 - receptacle)
30          (receptacleType ?x1 - receptacle ?x2 - itemtype)
31          (objectType ?x1 - obj ?x2 - itemtype)
32          (holds ?x1 - obj)
33          (holdsReceptacle ?x1 - receptacle)
34          (holdsAny)
35          (unknown ?x1 - location)
36          (explore)
37          (canToggle ?x1 - obj)
38          (isToggled ?x1 - obj)
39          (canSlice ?x1 - itemtype ?x2 - itemtype)
40          (isSliced ?x1 - obj)
41          (emptyObj ?x1 - obj)
42          (emptyReceptacle ?x1 - receptacle)
43      )
44
45      (:action explore_MoveAgent
46          :parameters (?agent0 - agent ?from0 - location
47              ?loc0 - location ?mov0 - movement)
48          :precondition (and (atLocation ?agent0 ?from0)
49              (move ?mov0) (conn ?from0 ?loc0 ?mov0)
50              (unknown ?loc0) (not (explore)))
51          :effect (and
52              (not (atLocation ?agent0 ?from0))
53              (atLocation ?agent0 ?loc0)
54              (not (unknown ?loc0))
55              (explore))
56      )
57
58      (:action MoveAgent
59          :parameters (?agent0 - agent ?from0 - location
60              ?loc0 - location ?mov0 - movement)
61          :precondition (and (atLocation ?agent0 ?from0)
62              (move ?mov0) (conn ?from0 ?loc0 ?mov0))
63          :effect (and
64              (not (atLocation ?agent0 ?from0))
65              (atLocation ?agent0 ?loc0))
66      )
67
68      (:action pickupObject
69          :parameters (?agent0 - agent ?loc0 - location ?obj0 - obj)
70          :precondition (and (atLocation ?agent0 ?loc0)
71              (objectAtLocation ?obj0 ?loc0)
72              (not (isInReceptacle ?obj0)) (not (holdsAny)))
73          :effect (and
74              (not (objectAtLocation ?obj0 ?loc0))
75              (holds ?obj0)
76              (holdsAny))
77      )
78
79      (:action pickupObjectFrom
80          :parameters (?agent0 - agent ?loc0 - location
81              ?obj0 - obj ?recep0 - receptacle)
82          :precondition (and (atLocation ?agent0 ?loc0)
83              (objectAtLocation ?obj0 ?loc0) (isInReceptacle ?obj0)
84              (inReceptacle ?obj0 ?recep0) (not (holdsAny)))
85          :effect (and
```

```
86            (not (objectAtLocation ?obj0 ?loc0))
87            (holds ?obj0)
88            (holdsAny)
89            (not (isInReceptacle ?obj0))
90            (not (inReceptacle ?obj0 ?recep0)))
91      )
92
93      (:action putonReceptacle
94          :parameters (?agent0 - agent ?loc0 - location ?obj0 - obj
95              ?otype0 - itemtype ?recep0 - receptacle
96              ?rtype0 - itemtype)
97          :precondition (and
98              (atLocation ?agent0 ?loc0)
99              (receptacleAtLocation ?recep0 ?loc0)
100             (canContain ?rtype0 ?otype0) (objectType ?obj0 ?otype0)
101             (receptacleType ?recep0 ?rtype0)
102             (holds ?obj0) (holdsAny) (not (openable ?rtype0))
103             (not (inReceptacle ?obj0 ?recep0)))
104         :effect (and
105             (not (holdsAny))
106             (not (holds ?obj0))
107             (isInReceptacle ?obj0)
108             (inReceptacle ?obj0 ?recep0)
109             (objectAtLocation ?obj0 ?loc0))
110     )
111
112     (:action putinReceptacle
113         :parameters (?agent0 - agent ?loc0 - location ?obj0 - obj
114             ?otype0 - itemtype ?recep0 - receptacle
115             ?rtype0 - itemtype)
116         :precondition (and
117             (atLocation ?agent0 ?loc0)
118             (receptacleAtLocation ?recep0 ?loc0)
119             (canContain ?rtype0 ?otype0) (objectType ?obj0 ?otype0)
120             (receptacleType ?recep0 ?rtype0) (holds ?obj0)
121             (holdsAny) (openable ?rtype0)
122             (not (inReceptacle ?obj0 ?recep0)))
123         :effect (and
124             (not (holdsAny))
125             (not (holds ?obj0))
126             (isInReceptacle ?obj0)
127             (inReceptacle ?obj0 ?recep0)
128             (objectAtLocation ?obj0 ?loc0))
129     )
130
131     (:action toggleOn
132         :parameters (?agent0 - agent ?loc0 - location ?obj0 - obj)
133         :precondition (and (atLocation ?agent0 ?loc0)
134             (objectAtLocation ?obj0 ?loc0) (not (isToggled ?obj0)))
135         :effect (and
136             (isToggled ?obj0))
137     )
138
139     (:action heatObject
140         :parameters (?agent0 - agent ?loc0 - location ?obj0 - obj
141             ?otype0 - itemtype ?recep0 - receptacle
142             ?rtype0 - itemtype)
143         :precondition (and
144             (atLocation ?agent0 ?loc0)
145             (receptacleAtLocation ?recep0 ?loc0)
146             (canHeat ?rtype0 ?otype0)
147             (objectType ?obj0 ?otype0)
148             (receptacleType ?recep0 ?rtype0)
149             (holds ?obj0) (holdsAny))
150         :effect (and
```

```
151            (isHeated ?obj0))
152        )
153 )
```

Next, Listings 2 shows the generated the planning problem based on the initial observation. This involves specifying the initial state of the environment and the goal that we want to achieve through the planning process.

Listing 2: ALFRED PDDL problem file with task achievement goal

```
1  (define (problem problem0)
2      (:domain alfred_task)
3
4      (:objects
5          agent0 - agent
6          AlarmClockType AppleType ArmChairType BackgroundType
7          BaseballBatType BasketBallType BathtubBasinType BathtubType
8          BedType BlindsType BookType BootsType BowlType BoxType
9              BreadType
9          ButterKnifeType CDType CabinetType CandleType CartType
10         CellPhoneType ... - itemtype
11         f0_0_0f f0_0_1f f0_0_2f f0_0_3f f0_1_0f - location
12         MoveAhead RotateLeft RotateRight - movement
13         empty0 - obj
14         emptyR - receptacle
15     )
16
17     (:init
18         (emptyObj empty0)
19         (emptyReceptacle emptyR)
20         (canContain BedType CellPhoneType)
21         (canContain CounterTopType PotatoType)
22         (canContain CoffeeTableType StatueType)
23         (canContain DiningTableType PanType)
24         ....
25         (openable MicrowaveType)
26         (atLocation agent0 f0_0_0f)
27         (move RotateRight)
28         (move RotateLeft)
29         (move MoveAhead)
30         (canHeat MicrowaveType PlateType)
31         (canHeat MicrowaveType PotatoType)
32         (canHeat MicrowaveType MugType)
33         (canHeat MicrowaveType CupType)
34         (canHeat MicrowaveType BreadType)
35         (canHeat MicrowaveType TomatoType)
36         (canHeat MicrowaveType AppleType)
37         (canHeat MicrowaveType EggType)
38         (conn f0_0_1f f0_0_0f RotateLeft)
39         (conn f0_0_3f f0_0_2f RotateLeft)
40         (conn f0_0_0f f0_0_1f RotateRight)
41         (conn f0_0_1f f0_0_2f RotateRight)
42         (conn f0_0_2f f0_0_3f RotateRight)
43         (conn f0_0_0f f0_1_0f MoveAhead)
44         (conn f0_0_2f f0_0_1f RotateLeft)
45         (conn f0_0_3f f0_0_0f RotateRight)
46         (conn f0_0_0f f0_0_3f RotateLeft)
47         (unknown f0_1_0f)
48     )
49
50     (:goal
51         (exists
52             (?goalObj - obj ?goalReceptacle - receptacle)
53             (and (inReceptacle ?goalObj ?goalReceptacle)
54                 (objectType ?goalObj PotatoType)
```

```
55                  (receptacleType ?goalReceptacle TableType)
56                  (isHeated ?goalObj)))
57       )
58 )
```

As not plan was found for the problem in Listing 2, the goal is changed to exploration in listing 3.

Listing 3: ALFRED PDDL problem file with exploration goal

```
1      (:goal
2          (and (explore ) (not (holdsAny )))
3      )
```

