# OpenReview forum: "Egocentric Planning for Scalable Embodied Task Achievement"
_NeurIPS.cc/2023/Conference — NeurIPS 2023 poster_

### Official Review · Reviewer_ZvXu · 2023-07-04

**Soundness:** 3 good
**Presentation:** 2 fair
**Contribution:** 3 good
**Rating:** 6
**Confidence:** 4

**Summary:**

The paper introduces Egocentric Planning, alternating exploration and task solving symbolic planning for long-horizon object-oriented POMDP in environments with deterministic action effects. The presented method is used as the planner in a hybrid agent, SOTA in the 2022 ALFRED benchmark, with neural perception. The agent is based on a previous successful design (FILM), with semantic SLAM in the latter replaced by a graph representing the current knowledge about the scene, besides the novel open loop planning algorithm exploiting this graph. The experimental results show a considerable improvement over FILM at the cost of longer trajectories required to succeed due to the time required to gather information about the scene, besides error analysis  and ablations. An exciting result is the generalization to new task types not present in training.

**Strengths:**

- The proposed method enables out-of-the-box generalization to new task types (within the same set of objects and relationships).
- The graph representation and planner algorithm work as a tailored alternative to semantic SLAM for object-oriented POMDPs with remarkable performance.

**Weaknesses:**

- The presented approach is effective under the assumption of deterministic action effects, but in real-world usage this should not generally apply. The paper consequentially discusses some possible ways to overcome this limitation, but also claims some better performing alternatives should be considered as baselines rather than comprehensive solutions, since they are limited in scope. I'd encourage rephrasing that discussion, which as is could be interpreted as unfair to the alternatives.
- The need for an exploration phase, when the egocentric planner is in principle designed to determine when to explore, seems a little ad hoc. I wish there was a clearer explanation why this cannot be provided by the main algorithm.

**Questions:**

1. How could the proposed framework be extended for multiple agents interacting with the environment?
2. Related to one of the weaknesses, what is missing in the egocentric planner to be capable of successfully exploring the environment until enough information is gathered to solve the task?

**Limitations:**

I think the paper makes a good job describing limitations of the proposed method, but, as mentioned in Weaknesses, I think the judgment of better performing alternatives as baselines, given the acknowledged limitations of the proposed method, could be seen as unfair, so I think an improved contextualization is needed.

---

> ### Author Rebuttal · Authors · 2023-08-10
>
> Thank you for your comments. We appreciate the insight into our work in the multi-agent setting.
>
> > **Strengths:**
> >
> > The primary contribution is that our method enables generalized solutions for new task types, using the same set of objects and relationships.
> >
> > **Weaknesses:**
> >
> > Your point on the effectiveness under deterministic action effects is taken. Our method is distinct from those with native support for non-deterministic actions. While the egocentric algorithm doesn't require deterministic actions, our implementation for ALFRED does. This iterative algorithm is our baseline for studying high generalization, even in deterministic domains.
> >
> > The exploration phase, as you noted, seems ad hoc. We don't propose a universal solution but applied a specific one for ALFRED, enough to win the competition. This approach is open to integration with other SLAM methods, offering a promising avenue for improvement.
> >
> > **Questions:**
> >
> > - "How could the proposed framework be extended for multiple agents?"
> >
> >   A subset of multi-agent problems can be mapped into classical planning and adapted to our method, as per Muise et al. (2015).
> >
> > - "What is missing in the egocentric planner to explore the environment until enough information is gathered?"
> >
> >   The egocentric planner's current optimization restricts exploration due to its limitation on actions and the imperfection of a learned Unet. Our initial steps of exploration help generate reliable candidates, but with improved depth perception and budget, the current setup could explore effectively. Future work may weight the graph with rewards associated with locations where objects can be perceived, guiding A* to those locations.
> >
> > **Limitations:**
> >
> > We acknowledge your thoughts on the judgment of alternatives as baselines and will revise the manuscript. Our focus is on generalization, whereas other methods may have different priorities.

---

> > ### Comment · Reviewer_ZvXu · 2023-08-14
> > **Thank you for your answers**
> >
> > In first place, I would like to thank the authors for their responses. I would also like to clarify the point of one of my questions.
> >
> > Q1. I was actually interested in a high-level description of the management of the information graph/mental state in each agent. For example, how would the state of objects that have undergone changes without the agent's interaction (e.g. caused by the environment, or by an external agent) be updated?

---

> > > ### Author Response · Authors · 2023-08-14
> > > **Managing Multi-Agent Interactions and State Updates**
> > >
> > > Thank you for the clarification. The central issue here pertains to the subset of changes that are relevant to the current plan. These changes might render actions inapplicable, causing the plan to fail in achieving its goal. To address this, the state can only be fixed through new observations or by communicating with other agents; however, let’s set aside the communication aspect for now.
> > >
> > > The notion of unbounded world modifications would make planning fundamentally impossible, necessitating some assumptions on our part.
> > >
> > > In the scenario of a bounded number of relevant world changes, it becomes straightforward to adapt the algorithm. Here, the state is updated after each action, and we can then inexpensively verify if the remainder of the current plan aligns with the current goal (e.g., using logical regression [1]), be it exploration or task achievement. Should this approach fail, we can replan. This simplistic idea might fall short in situations where changes lead to further exploration possibilities, like a corridor opening after moving furniture. In such a case, the agent won’t recognize the opportunity unless it revisits the location.
> > >
> > > When it comes to modifications made by other agents, we can design new egocentric planning algorithms drawing inspiration from existing Multi-Agent planning literature. A relevant survey, such as the one by [2] Torreno et al., “Cooperative multi-agent planning: A survey,” can provide valuable insights into the deterministic cooperative setting. An egocentric planning algorithm for multiple agents might be derived from one of these algorithms, as it assumes the underlying planner’s relies on full observability.
> > >
> > > To illustrate, let’s look at the FurnMove challenge [3], which operates in the same simulator as ALFRED. Agents must possess local information and policies, and in an algorithm centered on egocentric planning, each agent will maintain its map. Implicit coordination can be achieved if every agent plans for all agents but executes only its actions. This approach may mimic human coordination, where, knowing a piece of furniture must be moved, individuals instinctively grab the side closest to them.
> > >
> > > - [1] C. Fritz and S. McIlraith. Monitoring plan optimality during execution. In Proceedings of the 17th International Conference on Automated Planning and Scheduling (ICAPS-07), pages 144–151, 2007.
> > > - [2] Torreno, et al. “Cooperative multi-agent planning: A survey.” ACM Computing Surveys (CSUR) 50.6 (2017): 1-32. https://arxiv.org/abs/1711.09057
> > > - [3]. https://ai2thor.allenai.org/FurnMove/

---

> > > > ### Comment · Reviewer_ZvXu · 2023-08-14
> > > >
> > > > Thanks for the detailed response. When reading the original manuscript, given the dramatic performance drop without exploration phase, I concluded that most of the information about the environment is obtained during that initial phase, and then plans are essentially built on top of the quasi-static graph (with only occasional/minor exploration actually occurring during the Egocentric Planner execution).
> > > >
> > > > As mentioned in your answer, updating the knowledge graph/mental state to adapt to changes occurred without the agent's interaction might require a non-trivial amount of exploration, which the proposed method seems currently far from achieving without the explicit (and exhaustive) exploration phase. Is it correct that improving depth perception and removing budget restrictions would essentially suffice in providing the required level of exploration to perform reasonably in such setting? This is what I interpret from the first part of your answer.

---

> > > > > ### Author Response · Authors · 2023-08-14
> > > > >
> > > > > Thank you for the question. You are definitely correct. The reliability of depth perception and budget constraints are the main limitations preventing us from using our egocentric exploration planner from the very beginning. The initial exploration is used to generate a diverse set of clusters of objects in the environment for our agent to explore (for example, a table full of food and plates). Our current implementation tends to exhaustively search within a cluster of objects before moving on to another area, which could result in not finding the desired object within the movement budget. This is one of the major reasons for failure for ALFRED, as shown under "Object not found" in Table 2.
> > > > >
> > > > > This behavior is desired because interaction for ALFRED requires the agent to generate a segmentation mask. For small objects or overlapping objects, this is fairly problematic. We often found the need to explore different positions around an object to find the best candidate for interaction. Since we only have a budget of 10 failures, we have to be conservative in how we approach object interaction. In small environments, like an enclosed bathroom, we observe that we can often find the objects of interest during the initial exploration phase. However, in larger rooms with multiple object clusters, exploration after the initial phase using the planner is required for solving the task.

---

> > > > > > ### Comment · Reviewer_ZvXu · 2023-08-15
> > > > > > **Thanks**
> > > > > >
> > > > > > Thank you for the clarifications. Being my initial concerns cleared by a more detailed contextualization, which I hope will also be reflected in the final manuscript, I raise my rating.

---

> > > > > > > ### Author Response · Authors · 2023-08-15
> > > > > > >
> > > > > > > Thank you for the fruitful discussion and reconsideration! Your feedback has helped us clarify our contributions and how to improve the manuscript to make that clear. We are committed to further refining the presentation for greater clarity if accepted. Your time and perspective have made this possible. Thanks again!

---

### Official Review · Reviewer_JTSQ · 2023-07-06

**Soundness:** 3 good
**Presentation:** 3 good
**Contribution:** 3 good
**Rating:** 6
**Confidence:** 4

**Summary:**

The authors propose an approach combining symbolic planning and object-oriented POMDPs for symbolic planning, which gets extremely strong performance on the ALFRED benchmark and won the CVPR ALFRED challenge. Their approach uses PDDL, but extends it with a set of exploration-focused actions. They use a combination of explicit knowledge and heuristics (exploring close to a seen object for example) that are learned from data. They explore for 500 steps, using this to build a spatial-semantic graph (instead of an explicit map like many previous methods). They then use this with an off-the-shelf task planner to choose which actions to execute.

**Strengths:**

- Building a spatial semantic graph seems like a better (more scalable) way of solving ALFRED tasks than explicit maps
- Strong performance on a well-respected benchmark
- A lot of great ideas, and some of the explanation is very good
- Great to see a *new* approach to solving ALFRED, not just building off of FILM/HLSM

**Weaknesses:**

- The name "egocentric planning" doesn't make a lot of sense to me; everything in ALFRED is going to be some manner of egocentric planning
- Not clear how general this is - lots of engineering in the planning domain. More analysis would help here; ideally the authors could run on a different domain (OVMM might be one - ovmm.github.io), but there really aren't good options with strong existing baselines like ALFRED. Instead maybe they could describe in more detail how it would be applied to other domains.
- Writing could be improved. Several typos (Apporach --> Approach), IER (exploration algorithm) not being referenced by name in text.
- It's not really clear this is a *learning* contribution - learning components seem minor. I think this is ok, because it's still a useful result on a learning problem, but I could see the argument against it.
- Very engineering heavy.

**Questions:**

- How is the spatial graph constructed? It seems like a lot of the detail here is lacking.
- How important are the 500 exploration steps? Seems like it should be able to work well without this, if it's really a good POMDP planner.
- Construction of the initial state was also a bit unclear to me.

**Limitations:**

- Only applied to one benchmark
- Need for 500 exploration steps at the beginning seems weird
- Very dependent on perception models - perfect detection, depth, etc., which limits application to other domains.

---

> ### Author Rebuttal · Authors · 2023-08-10
>
> Thank you for your comments. We've addressed some of them in the general response.
>
> > **Strengths:**
> >
> > - A lot of great ideas; some explanations are very good
> > - Great to see a new approach to solving ALFRED, not just building off FILM/HLSM
>
> We look forward to elaborating on these ideas.
>
> > **Weaknesses:**
>
> > - "Egocentric planning" doesn't make sense; everything in ALFRED is egocentric planning
>
> Egocentric planning emphasizes using a planner with full-observability for problems under partial observability. We intend to keep working in this direction. What would you suggest as a name?
>
> > - Unclear generality; lots of engineering in the planning domain
>
> See the general response for a discussion on planning domains. Our method emphasizes scalability and could be a target for learning.
>
> > - More analysis needed; describe how it would apply to other domains
>
> OVMM is a possible domain, but physical manipulation is beyond our scope.
>
> > - Writing could be improved
>
> Thank you. We'll correct typos and inconsistencies.
>
> > - Not clear this is a learning contribution
>
> Our focus is on integrating existing models. While perceptual models improve, multi-step generalization remains challenging.
>
> > - Very engineering heavy
>
> The task required initial domain engineering to create a set of action object types. The planning actions should generalize to objects beyond ALFRED.
>
> - Generic: Perceptual models, Egocentric algorithm
> - Environment: actions, object types
> - Engineering: PDDL model, Semantic part updating
> - ALFRED tuning: exploration, policy changes
>
> > **Questions:**
>
> > - How is the spatial graph constructed?
>
> See general response.
>
> > - Importance of 500 exploration steps?
>
> Ours is a lightweight POMDP solver. Early exploration is less useful. SLAM methods could be used to update semantic maps.
>
> > - Construction of the initial state?
>
> The initial state comprises a location and perceived objects in the facing direction.
>
> > **Limitations:**
>
> > - Only applied to one benchmark
>
> Indeed, but our method might enable other applications with trained models for the AI2Thor environment.
>
> > - Very dependent on perception models
>
> Our contribution abstracts the perception model, widening applicability as generalist models improve.

---

### Official Review · Reviewer_vi1s · 2023-07-06

**Soundness:** 3 good
**Presentation:** 2 fair
**Contribution:** 3 good
**Rating:** 4
**Confidence:** 3

**Summary:**


The paper presents a modular approach that combines symbolic planning and object-centric Cost POMDP for solving ALFRED tasks. The proposed method demonstrates improvements compared to previous end-to-end and modular approaches, such as FiLM. Unlike methods like FILM, HLSM, and Prompter, EPA utilizes a semantic graph representation instead of a top-down 2D map.

The method incorporates an initial phase of 500 exploration steps to gather sufficient knowledge, which is crucial for determining an appropriate expandable initial state. Object information is selectively saved only when objects are within immediate reach (0.25m ahead). This selective saving facilitates the conversion of observations into a symbolic state and reduces the length of generated plans.

The authors' findings suggest that the performance enhancements achieved by EPA over FILM and similar approaches primarily stem from its iterative planning approach. This approach enables the agent to recover from failure scenarios through flexible subgoal ordering, leading to improved performance.

According to the authors, EPA achieved the second-highest ranking on the ALFRED leaderboard, closely following the Prompter method. The superiority of Prompter, however, is attributed to modifications in obstacle size and reachable distance rather than the use of prompts.

**Strengths:**

1. The paper demonstrates the use of preconditions and effects through PDDL to improve the overall success of long-horizon tasks, especially unseen success rate.
2. The paper combines the use of symbolic planning using learned vision and language models and highlights how certain aspects of generalization can be achieved by abstraction.

**Weaknesses:**

1. The current assumptions on semantic spatial graphs require random exploration for 500 steps to visit each location and form a node in the graph. The paper reports a drop in performance if this initial observation phase is ignored. This approach has two major limitations: (1) it dramatically increases the timestep overhead for the proposed agent as compared to the existing works. (2) it assumes a static unchanging environment after mapping and will likely fail in realistic environments with dynamic obstacles. Given the existing visual-inertial SLAM approaches, as noted by the authors, it seems that this issue can be mitigated. Some existing approaches for topological mapping [1] might also be relevant, and in turn, improve the path length weighted success rate (PLWSR).

1. Writing PDDL domain definitions and problems for each task is known to be a tedious coding task. Any errors in representing the available objects and actions would yield no plan. The current approach seems too close to reverse engineering the process of creating trajectories for the ALFRED task.  While the authors report 150 hours for PDDL domain and problem definitions, a large chunk of the work involving object types and action predicates is already described in the ALFRED metadata. This does not give a reasonable perspective on how many hours would it take to scale and maintain this approach further, especially in the physical world.

 *[1] Chaplot, D.S., Salakhutdinov, R., Gupta, A. and Gupta, S. 2020. Neural Topological SLAM for Visual Navigation. In CVPR.*


**Questions:**

1. Could you kindly provide some clarification regarding the semantic spatial graph? Specifically, what information does it contain and what is the average size of such a graph? The description in Appendix F2 suggests that the graph incorporates visual observations, segmentation, and depth. I would like to understand if this graph represents the visual scene "in front of" the agent. If so, I have a couple of related questions:
    1. In case the agent is at the same location but facing a different direction, would the information on the graph be overridden?
    2. If not, does each node in the graph contain a representation of the "360-degree visual observation" at that particular location? How are the "actions" represented as edges in this context?
2. I'm curious to know how the approach encodes common sense knowledge or visual knowledge (mentioned in Lines 57-58) as part of the domain definition in PDDL.
3. Could you please explain how the "exploration actions" are defined? In Appendix F3, Listing 3, it is mentioned that the agent cannot hold something and explore the environment. For example, can the agent actively search for the coffee after picking up the cup? I would appreciate some clarification on this.
4. I would like to understand how the possible predicates are listed in the PDDL. Additionally, how are the preconditions and effects identified for each predicate? Are they learned or inferred from visual observations, language goals, or the agent's interaction to gather information? Are these predicates hard-coded by a human? If so, I'm interested in understanding how this differs from the classical task planning setup.
5. It is not entirely clear how the generalization to new tasks is achieved, particularly when the language module is trained to output a task type out of seven tasks (as mentioned in lines 73-74). Could you please elaborate on the meaning of the statement in lines 308-309, "Our egocentric planning breaks a task into a set of goal states, autonomously generating an action sequence"?

**Limitations:**

Overall, it appears that the current approach is heavily focused on reverse engineering the trajectory creation process for the ALFRED task, utilizing learned vision and language models. While the method demonstrates a high success rate on the ALFRED benchmark for unseen scenarios, there is room for improvement in terms of clarity and the overall significance of the proposed approach. It remains uncertain how applicable the use of semantic spatial graphs or PDDL domain definitions would be beyond the ALFRED simulated benchmark, particularly when considering real-world physical environments.

---

> ### Author Rebuttal · Authors · 2023-08-10
>
> Thank you for your comments. Some of your comments were addressed in the general response.
>
> ## Strengths
>
> > The paper demonstrates:
> > - The use of preconditions and effects through PDDL to improve the overall success of long-horizon tasks, especially unseen success rate.
> > - The combination of symbolic planning using learned vision and language models, highlighting how generalization can be achieved by abstraction.
>
> Those were part of our goals. Thank you.
>
> ## Weaknesses
>
> > - The current assumptions on semantic spatial graphs require random exploration for 500 steps to visit each location and form a node in the graph.
> > - Writing PDDL domain definitions and problems for each task is tedious. While 150 hours were reported for PDDL domain and problem definitions, scaling and maintaining this approach's perspective is not clear, especially in the physical world.
>
> See general response on PDDL construction and semantic spatial graphs.
>
> > [1] Chaplot, D.S., Salakhutdinov, R., Gupta, A. and Gupta, S. 2020. Neural Topological SLAM for Visual Navigation. In CVPR.
>
> Thank you. We will add the citation. Very relevant.
>
> ## Questions
>
> ### Semantic Spatial Graph
>
> - **Information and Size:** The semantic map contains visual and depth information. It stores actionable items, their class, average pixel distance, obstacles, and more. The average size of such a graph is around 500 from our observation.
> - **Different Directions:** We encode unique nodes with different orientations, and there is no 360-degree observation. A detailed explanation of how the coordinates and edges are handled is provided.
>
> ### Domain Definition in PDDL
>
> - **Common Sense and Visual Knowledge Encoding:** Extra goals can be added in PDDL, and LLM can help turn comments into extra PDDL goals.
> - **Predicates and Preconditions:** Planning domains are available, human-written, or learnable. Anchor types are identified, and an egocentric algorithm solves planning problems centered on the agent location.
> - **Generalization to New Tasks:** We tested on new tasks written directly as PDDL goals.
>
> ### Exploration Actions
>
> - Exploration actions are defined to move to unexplored locations. They update the spatial graph and are affected by holding objects that obscure the view.
>
> ### Generalization and Task Handling
>
> - **Language Module and Generalization:** Elaboration on how the egocentric planning breaks a task into goal states, autonomously generating an action sequence.
> - **Applicability and Reverse Engineering Concerns:** The approach focuses on ALFRED but generalizes to objects and planning actions beyond it. The reverse engineering concern and the point of ALFRED in generalization are addressed.
>
> ## Limitations
>
> - **Clarity and Significance:** Improvement in terms of clarity and the overall significance of the proposed approach is needed.
> - **Real-world Applicability:** Uncertainty regarding the applicability of semantic spatial graphs or PDDL domain definitions beyond the ALFRED simulated benchmark, particularly in real-world physical environments.
>
> Answer to physical "environments" will be provided in the general response.

---

> > ### Author Response · Authors · 2023-08-15
> >
> > Thank you for your thoughtful questions. In response to your specific query about how our approach differs from the classical task planning setup, we would like to add:
> >
> > We employ a classical planning model for ALFRED, but apply it to problems under partial observability where new objects may be revealed, both features beyond classical planning. The key advantage is that classical planning models are simpler to create. Our innovation lies in exploiting anchor predicates and exploration actions without requiring a more complex planning model.
> >
> > Please let us know if there's any further clarification we can provide on the concerns you raised, and thank you again for looking into what we've accomplished.

---

### Official Review · Reviewer_dRsd · 2023-07-06

**Soundness:** 4 excellent
**Presentation:** 4 excellent
**Contribution:** 3 good
**Rating:** 7
**Confidence:** 4

**Summary:**

This paper studies the problem of embodied tasks in which the agent needs to plan over long task horizons given natural language instruction. Current methods that use end-to-end training leads to entangled representation, which makes it hard to solve the task. On the other hand, planning methods such as PDDL can produce high-quality actions given a well-defined problem specification. To solve the task, the paper proposes a method that consists of two parts: (1) goal-oriented exploration that aims to gather missing information, and (2) a classical planning method that aims to produce a feasible action. The proposed method is evaluated on the embodied benchmark ALFRED. The method improves the prior SOTA performance by 8.3%, winning the ALFRED challenge at CVPR 2022 Embodied AI workshop.

To be more precise, the proposed method consists of several parts: (1) a visual module for semantic segmentation and depth estimation of the scene, (2) an egocentric planner for planning given the information gather, (3) a semantic spatial graph for scene memorization. At the beginning of the task, the agent is provided with 500 steps to explore its surroundings. After that, the information is converted to a semantic spatial graph for the input of PDDL. Figure 1 shows the method. The algorithm inerates using the following steps: (1) find a path for reaching the goal, (2) if the goal is not reached, do exploration again

Table 1 shows the main result of the paper. And the rest of the experiments section provides a detailed ablation study of the method.

**Strengths:**

1. The writing of the paper is clear and easy to follow. For instance, in the introduction, I can easily understand the motivation of the method.

2. The proposed method is a winning approach at the CVPR Embodied AI workshop in 2022. This shows that the method has been thoroughly tested, and the result is convincing.

3. In the AI era, it is nice to see a classical planning method is robust in such tasks over a neural network-based method.

4. The related work is reasonable. It covers important papers such as Saycan

5. Overall, I think this is a good paper, and the method is elegant and promising.

**Weaknesses:**

The proposed method could be very specific to the ALFRED task, in the sense that the method is solely optimized and engineered for the  ALFRED task. For example, in some tasks, there is no abstraction of the action space, like OpenDrawer.

**Questions:**

1. If I want to deploy such a method in the real world, what would be the challenge and the additional step to do this?

2. Following the previous questions, what is the Sim-to-real gap here?

3. Will the method be able to handle tasks such as you need to open the drawer to find the coffee mug in it, instead of target objects being visible to the agent? I will say this is more challenging in the POMDP setting than the setting in the paper.

4. What are some failure cases? Could you provide a couple of examples?

5. Is it possible to integrate such a method with semantic exploration (https://devendrachaplot.github.io/projects/semantic-exploration)?

**Limitations:**

The paper has clearly stated the limitations of the paper, including the method cannot recover from irreversible failures, being sensitive to perception errors, and having no memory for belief tracking.

---

> ### Author Rebuttal · Authors · 2023-08-10
>
> Thank you for your comments. The connection with SayCan is interesting, as our work complements theirs.
>
> > **Weaknesses:**
>
> > The proposed method might be too specific to the ALFRED task, in the sense that the method is optimized solely for this task. For example, there's no abstraction of the action space, like OpenDrawer.
>
> We agree that competitions—like metrics—often lead to specific optimizations. We tuned parameters, such as random exploration and object reachability depth. Yet, some optimizations are principled methods, like using graph reachability to eliminate irrelevant actions. This method is applicable beyond the ALFRED challenge.
>
> > If I want to deploy this method in the real world, what would be the challenges and additional steps?
>
> Our approach targets errors at the action level. For any application, ensure that actions perform as intended. In realistic environments, it is prudent to guarantee the robot's capabilities. Refer to our planning-based modular printer example.
>
> > What is the Sim-to-real gap here?
>
> Robust actions foster high accuracy. Ruml et al., JAIR 2011, describe a real-time modular printer. [Here's the video](https://www.jair.org/index.php/jair/article/view/10693). Noisy actions or unexpected results are outside our scope, but limited support is offered for failed actions.
>
> > Will the method handle tasks like opening a drawer to find a coffee mug?
>
> While we only support deterministic actions, classical planning can tackle partial observability. By following Albore, Palacios, Geffner, we can create hypothetical objects and optimistic actions. Upon interaction, if a mug is not found, the agent will seek it elsewhere.
>
> > What are some failure cases?
>
> Failures often arise from object recognition and distance estimation errors. An agent might misidentify an apple or collide with an obstacle. These failures are detailed in Table 2 on page 7.
>
> > Is it possible to integrate this method with semantic exploration?
>
> Thank you for the reference. Our plans can play the role of the local policy $\pi_L$ in SemExp, and we can integrate with other SLAM algorithms by using SLAM updates instead of our own.
>
> > **Limitations:**
>
> > The paper states the limitations, including irrecoverable failures, sensitivity to perception errors, and no memory for belief tracking.
>
> Thank you. We do perform belief tracking in ALFRED. The semantic graph contains current information, and unvisited locations implicitly represent unknown objects and locations. Our method's open-world nature means we don't assume a finite number of objects. This relates to the 0-approximation notion by [Baral et al., AIJ 2000](https://www.sciencedirect.com/science/article/pii/S0004370200000436).

---

### Official Review · Reviewer_oR6u · 2023-07-10

**Soundness:** 3 good
**Presentation:** 2 fair
**Contribution:** 3 good
**Rating:** 5
**Confidence:** 4

**Summary:**

The authors propose a hybrid approach leveraging neural perception models and symbolic planners for egocentric planning and task completion in embodied environments. They demonstrate their approach in ALFRED benchmark winning the 2022 CVPR challenge. Their central idea is the use of symbolic planners, which are typically used in fully observable settings. To deal with partial observability, they perform iterative exploration with symbolic planning. Structurally, their implementation is similar to the previous SOTA on ALFRED — FILM. However, instead of using a semantic map as FILM does, they use a graph structure for object and location information storage. This graph is updated through exploration and then used by the downstream symbolic planner.

**Strengths:**

- The authors’ hybrid approach leveraging symbolic planners is novel amongst the modular approaches for long-horizon EAI tasks, and can be promising for future research.
- The authors won the CVPR 2022 EAI ALFRED challenge.

**Weaknesses:**

- Poor clarity of exposition: I found the major aspects of authors’ technical approach poorly explained in the paper. This really limits reproducibility as well as use by the community IMO and consequently limits the value of the paper.
    - Spatial graph: Unclear how the graph is built and what does it look like. The authors say that the graph encodes location as the node key and visual observations as values. However, it is unclear what is the co-ordinate system for location (consequently would we get the same graph for the same environment and task but different agent start position?), what happens when the agent visits the location more than once, and lastly, how is it used for the low-level policies that have to navigate given there is no map.
    - Object-centric POMDP: The authors briefly mention the Cost POMDP but do not appropriately define or explain it at all in the main paper. Most definitions are pushed to supplementary. While it is okay to push nitty-gritty details to supplementary, without properly framing the problem and then grounding their approach in the framework, I am not sure if I can buy the authors’ claim that their approach is theoretically sound (L121).
    - Exploration: Unclear how exploration is handled. The only description is in the intro (L52-56).
    - The algorithm is not commented and contains symbols which are never defined or explained, making it difficult to understand what did the authors actually implement and why:
        - Symbol Ge is never defined
        - Unclear how cost c of action is computed and used in function $M^{PD}$.
    - The authors say that the PDDL domain and environment definitions of actions are misaligned, which make sense (L149). However they never explain how this alignment is obtained in their approach.
    - Furthermore, given the PDDL and planning terminologies might not be accessible/known to everyone, I encourage the authors to be more careful and clear in their descriptions. For instance, what exactly are anchor object types and why do we need them? Also unclear what other anchor object types can exist in EAI settings.
    - I didn’t get much out of Sec.4 and 5 despite the fact that I work with symbolic planning and EAI and even when there is so much to explain about the approach as mentioned above. I recommend rewriting entire Sec.4 and 5.
- Generalization: Authors claim and attempt to show generalization in Sec.8.4. However, this section had no details whatsoever about the new tasks they use nor a reference to supplementary. I managed to find some details in supplementary. However, I am sure a general reader will be lost here. Furthermore, the authors claim that they do better on the new tasks and achieve zero-shot success 82% times, better than other methods but do not show any such comparison in their work (tab.6 supplementary). Given this is one of the main claims of advantage of their approach, I’d like to see comparison with neural and template-based approachers, and also with the prompter method.

With the above (also see limitations section below), IMO the paper is not yet at par for publication at Neurips. In its current state, it can be a great workshop paper but requires major rewriting and additional comparisons e.g., for generalization claims otherwise.

**Questions:**

- Table1: main results should have bolded numbers and indications on which metrics need to be higher/lower for better task performance (e.g., with $\uparrow$ and $\downarrow$) to improve interpretability of results. Similarly sorting rows in Tab2 based on performance in unseen might be useful.
- L166: “This setup promotes more robust action sequences and generalization beyond the seven ALFRED-defined tasks” — not sure how?
- Results: Given that the LGS-RPA also comes close in terms of GC and SR (unseen) to the authors approach, I encourage the authors to discuss LGS-RPA in the result section as well.
- Tab:3 Unclear why the method only achieves ~60-70% GC with all ground truth available. Is this the upper bound? Why is the upper bound not 100% given that the env. is deterministic and we are using a symbolic planner?
- Is the planning performed from scratch every time? I imagine that the state of the world and thus the graph have incremental changes so wondering if the authors do anything smart to reduce the planning time at each iteration.

**Limitations:**

- The authors do not touch upon the issue of obtaining PDDL domain in the first place, which is core to their approach. Unclear how this would scale for real world applications/agents and how errors in domain file can impact their approach. Lastly, they mention that they can handle additional constraints e.g., energy during planning. However, in practice, I’ve found planning time to blow up when using durative and cost constraints. Unclear again how authors approach would scale for more complex problems. Also, on that note, I’d like to see planner time numbers for the tasks perhaps in the supplementary.
- Similarly, the authors say that they are more robust to perception errors however they say they do so using hand-crafted policies for replanning (Sec.8.3). Unclear if the robustness is because of these handcrafted policies or because of their iterative planning approach combining exploration and symbolic planning.
- The  section on broader impact and societal impact is missing. I'd encourage the authors to think about real world applications that their work might enable for this section.

---

> ### Author Rebuttal · Authors · 2023-08-10
>
> We thank the reviewer for their insightful review.
> There are indeed issues in the manuscript.
> We apologize for the lack of clarity.
> As we stated, we have already improved part of the issues in the manuscript and will certainly further improve it thanks to your feedback.
>
> The general response addresses the following issues mentioned in this review:
>
> - Object-oriented POMDP and framing of the problem.
> - Spatial semantic graph.
> - Misalignment between the environment and the symbolic model.
> - The PDDL provided by the user.
>
> Here, we provide additional answers:
>
> > **Spatial graph**: See the attached PDF and the general response.
> The coordinates are (x-coordinate, y-coordinate, facing direction).
> If the agent visits the same location more than once and faces the same direction as before, that will not create a new node.
> Visiting the same room twice would create the same graph if we fixed the random seed for exploration, and all other components were deterministic.
> If we let the agent explore the whole space in another order, the new resulting semantic graph would be equivalent to the previous one, except for object names created on the fly.
>
> > **Algorithm issue**: Apologies.
> The algorithm has been fixed.
> For instance, $G_e$ is now part of the Task that the algorithm receives as input.
> We define Tasks in the shared response.
>
> > **Cost computation**: The generic algorithm supports a planner that minimizes or reduces cost.
> The classical planner used in ALFRED does not use cost, but the plans tend to be short, as the domains are simple from the planning point of view.
>
> > **PDDL terminology**: We agree it might not be accessible.
> We will improve the manuscript by providing timely examples and leave the formal definitions for the supplementary material.
>
> > **Generalization claim**: We fixed the references and improved the description to emphasize that we are solving new tasks, starting with a symbolic goal.
> So, the 82% success rate is related to the 52.29% for w/ gt language in Table 3, valid unseen.
> The comparison is not meaningful as these are different tasks.
>
> > **Comparisons**: Implementing the new tasks in Prompter and FILM++ (Inoue et al 2022), and FILM would require retraining the classifier with new tasks and creating templates by hand.
> While our results rely on creating a symbolic planning model, its compositionality allows us to request the agent with unforeseen tasks.
> The best course of action depends on the domain.
>
> > **Publication readiness**: We hope this response has clarified how we will address the limitations of the current manuscript.
> Please let us know if there is anything else that we could address, which might change your assessment regarding our submission’s acceptance.
>
> > **Table 1 and 2 recommendations**: Will do.
> For all the metrics, higher is better.
> Success Rate demands to achieve the goal, while Goal-condition Success focuses on subgoals.
> However, in planning, achieving subgoals independently might not predict the success rate and have low value for the user.
>
> > **Clarification on robust action sequences**: We intended to say that our setup can solve tasks beyond the seven included in the ALFRED challenge.
> We will improve Section 8.4 to make this connection clear.
> The robustness of new tasks depends on the robustness of each action that might have been used in previous tasks.
>
> > **Anchor object types**: We discussed them in the general response.
> An anchor type in an extension of ALFRED could be a panel that shows the power status of appliances.
>
> > **LGS-RPA comparison**: LGS-PRA is a very interesting piece of work.
> However, we believe that the improved accuracy is due to techniques on landmark detection method and local pose adjustment, which are not the emphasis of our work.
> We find it difficult to conduct a proper comparison.
>
> > **Tab. 3 and upper bound**: There are two main reasons we cannot achieve 100% even with ground truth.
> First, the simulator can get stuck when holding large objects close to a wall, and some objects spawn in the corner cannot be reached due to fixed 90% turns.
> Secondly, our method does not build a graph for ‘turning up’ and ‘turning down’ actions, making some things unreachable.
>
> > **Planning process**: As classical planners expect a fixed set of objects, we plan from scratch every time we reach an exploration goal that might reveal new objects, or if there is an execution failure that we use for revising the semantic graph.
>
> > **PDDL scaling and constraints**: We discuss the obtention of PDDL in the general response.
> Writing, learning, or debugging PDDL leads to higher generalization, and the right decision depends on the domain.
> Regarding constraints, some can be mapped into classical planning with costs.
>
> > **Planner time numbers**: In the beginning, replanning is frequent but fast.
> When the graph is large enough, the plan can take up to 5 seconds but often takes around 1 second.
>
> > **Robustness**: Navigating around objects is a weak point of our approach.
> Our hand-crafted policy doubles down in this direction, following the ALFRED environment’s simplification.
> The strength of our approach lies in exploiting skills and perception across tasks.
>
> > **Broader and societal impact**: Thank you.
> We will elaborate on the potential benefits of our approach in this section, including robustness and higher explainability.

---

> > ### Comment · Reviewer_oR6u · 2023-08-21
> > **Thank you for the detailed rebuttal**
> >
> > Thanks for providing various clarifications, a graphic for the graph, and updating symbols and descriptions.  I am a little concerned that many technical descriptions in the paper have been changed but not reviewed fully (as in initial reviews), but I'll let the ACs decide if it would be okay to accept the paper in such a case. Either way, I'll raise my rating to "weak/borderline accept" -- I will not argue for the paper but I won't push back if other reviewers are supporting the paper/there is a champion for the paper.
> >
> > - It would be good to provide clear distinction on LGS-RPA and why a comparison isn't possible in the main paper. Same for the tasks used for generalization, to explain why other baselines are not possible.
> > - I am also concerned about the planning time numbers for real-world applications, perhaps the authors can add that in their discussions/limitation section.
> > - Also glad to see more clarity on PDDL assumptions and prior work on learning PDDL. Hoping authors can do the same in the main paper.

---

> > > ### Author Response · Authors · 2023-08-22
> > >
> > > Thank you for diving back in and being amenable to raising the score! All three suggestions would certainly make for a stronger paper, and we'll gladly make the additions to the final paper. Do let us know if you'd like us to surface any of that discussion here for further review, and thanks again for your insight!

---

### Author Rebuttal · Authors · 2023-08-10

## Introduction

We would like to extend our heartfelt gratitude to the reviewers for their insightful comments and constructive criticism. The feedback has been encouraging and instrumental in helping us understand the significance of our work in the following areas:

- Our **modular approach** (R1, R3)
- The **role of symbolic planning** in solving a diverse set of tasks (R1, R2, R3)
- Our unique **notion of the semantic graph** (R4)
- The success we demonstrated in our entry in the **ALFRED competition** (all reviewers)

However, the reviewers have raised certain questions regarding the semantic graph, the symbolic planning model, and their relationship. We appreciate the opportunity to respond and provide clarity on these aspects.

## Core Contributions

Before diving into the detailed response, we wish to revisit the core of our contribution. Our focus is on environments with fixed, known types, relationships, and actions grounded on them. We aim to solve a series of tasks in the same environment, and our winning entry in the ALFRED challenge is a testament to the method that allows us to explain a particular implementation in depth.

### The Essence of Our Method

- We present a lightweight method for achieving goals that can be expressed and solved in the fixed environment.
- In ALFRED, we reuse existing perceptual models and employ a simple exploration strategy, so the novelty of our method resides in the planning part.
- While ALFRED features only seven classes of tasks, our method supports others as long as natural language processing can map the user goal into the fixed object types and relationships.
- We expect non-symbolic planning methods to perform better with more data or by focusing on specific tasks. However, our method serves as a simple baseline for studying systematic generalization and compositionality per domain.

### The ALFRED Challenge

- Our method's effectiveness is shown by our success in the ALFRED challenge.
- Though limited to seven task classes, our approach can support other tasks.
- We demonstrate additional tasks that can be solved in the implicit Object-Oriented POMDP, consisting of the ALFRED environment and standalone models for object detection and semantic parsing of the tasks.

### Object-Oriented POMDPs

R1 rightly pointed out issues in sections 4 and 5 of our manuscript describing Object-Oriented POMDPs and the core algorithm. In response:

- We have massively rewritten these sections.
- The definition of Object-Oriented POMDP is now more finely defined.
- In section 5, we define an Environment as a tuple \(\mathcal{E} = \langle A_\mathcal{E}, \mathcal{T_\mathcal{E}}, \mathcal{V_\mathcal{E}}, \mathsf{reset}, \mathsf{step} \rangle\), with various elements representing the set of parameterized actions, object types, initial observation, and action execution.
- Our method's goal is to solve a series of tasks in the same environment, which we define as a Task, \(T_\mathcal{E} = \langle I_\mathcal{E}, G_\mathcal{E} \rangle\), setting the initial state and goal of the environment.

### Observations and Planning

Our approach entails three major components:

1. **Object-Oriented Environment**: The environment encapsulates object types, potential values, and actions.
2. **Egocentric Planning Algorithm**: This algorithm uses deterministic actions and re-planning, focusing on a fragment of Object-Oriented POMDPs where the set of objects is initially unknown.
3. **Symbolic Planning Model**: Assumed to be provided by the user, the symbolic model may be non-trivial but can be argued to be necessary for generalization.

We also introduce the notion of **anchor objects**, which may reveal the existence of new objects. This concept generalizes to other domains, allowing the agent to discover new options while navigating different environments, such as a warehouse or a web application.

### Alignment between Symbolic Model and Environment

The alignment between the symbolic model and the underlying environment can be challenging. However, in ALFRED, this alignment is facilitated by a spatial semantic map. For each unique environment, we use a spatial graph to encode the agent's location and direction, with movements becoming the edges. This simplifies navigation and serves as a top-down map.

### Engineering Decisions

Some reviewers raised questions about our engineering decisions in ALFRED, like the exploration strategy. We will respond to these directly but emphasize that our ALFRED entry demonstrated our approach successfully.

## Additional Considerations

- **500 Steps**: We need to address the choice of limiting actions to 500 steps, explaining the reasoning and potential outcomes if this constraint is not used.
- **Sim2Real Limitations**: Our approach does not support continuous actions or complex manipulation. We assume discrete actions.
- **Perception and Multimodal Models**: We utilize multimodal models such as CLIP [14] and DETIC [26] for open-vocabulary object detection.
- **Potential Applications**: Our method could be applied in various scenarios, as described in table 1 of "HomeRobot: Open-Vocabulary Mobile Manipulation."

## Conclusion

Our response aims to shed light on our method's novelty, the alignment between the symbolic model and environment, and other specific questions raised by the reviewers. We hope to have provided a comprehensive overview of our work and look forward to further discussions and feedback. Thank you again for your valuable insights.

---

> ### Author Response · Authors · 2023-08-11
> **Formulas format for revisions in sections 4 and 5**
>
> We realized that the format of the formulas did not work properly in the response. This revised format may clarify the visualization:
>
> - In the revised section 4, we define an *Environment* as a tuple:
>   $\mathcal{E} = \langle A_\mathcal{E}, \mathcal{T_\mathcal{E}}, \mathcal{V_\mathcal{E}}, \mathsf{reset}, \mathsf{step} \rangle$,
>   with corresponding definitions for each term.
> - A *Task* is defined as a tuple:
>   $T_\mathcal{E} = \langle I_\mathcal{E}, G_\mathcal{E} \rangle$,
>   setting the initial state and the goal through the $\mathsf{reset}$ function.
> - A new task can be considered a new initial state in the fixed Object-oriented POMDP, including hidden variables to express goals and user preferences.
> - In the revised section 5, Algorithm 1 receives as input:
> 	- Environment $\mathcal{E}$
> 	- Planning domain $\mathcal{PD}$, provided by the user
> 	- Anchor types and corresponding exploration actions, provided by the user
> 	- Agent’s mental state initialization function $\mathcal{M}^I_{\mathcal{PD}}$, mapping a task and initial perception into a belief state
> 	- Agent’s mental state update function $\mathcal{M}^U_{\mathcal{PD}}$, updating the agent's state and returning new planning objects and a modified initial state for the replanning loop
> - Algorithm 1 returns a successful trace or Failure if exploration failed and no plan to the symbolic goal was achieved.
> - For ALFRED, the mental state initialization and update create and refine the spatial semantic map.
> - Our contribution is a method for solving a series of tasks $T_\mathcal{E}$ for the same environment $\mathcal{E}$.
> - While the ALFRED challenge includes only seven task templates, our method generalizes to other tasks within the same environment. So, our method is more appealing when users require more flexible generalization.

---

> ### Comment · Area_Chair_7bAP · 2023-08-21
> **Thanks for detailed rebuttal**
>
> Thanks to the authors for individual responses to the reviewers and the general response. These span writing clarifications and fixes, generalization claims, sim-to-real gap, PDDL models and domain definition, assumption of deterministic state transitions, and specificity of contribution to ALFRED benchmark. The committee shall discuss further and take these into consideration.

---

### Author Response · Authors · 2023-08-11
**PDDL symbolic models**

We thank the reviewers for their insightful comments and questions. We hope we have addressed them satisfactorily. If there are any remaining concerns, please do not hesitate to let us know.

Regarding the authoring of the PDDL symbolic models, we understand that obtaining a symbolic model might be considered non-trivial. It's essential to differentiate the convenience of representing actions as PDDL symbolic models from the process of obtaining them.

We included a reference to [AKFM22], which explores learning to plan given pixel states and unlabeled actions. The paper utilizes a sequence of variational auto-encoders for learning an action model, demonstrating that symbolic composable action models contribute to high generalization.

For our context, learning a PDDL is relatively simpler since we assume that the environment includes object types and actions. We've added a citation to the review [AFP+18], which focuses on learning planning action models.

In our submission, we assume that the user provides a symbolic planning model with full observability. For ALFRED, we further assume that actions are deterministic. This simplifies the problem into classical planning, enabling the use of various planners and tools.

- Symbolic planners are available in  ([Planutils](https://github.com/AI-Planning/planutils)). [MPSK22]
- There is a [VSCode extension for PDDL](https://marketplace.visualstudio.com/items?itemName=jan-dolejsi.pddl) that has been actively developed since 2017.

Moreover, recent advancements in pre-trained large language models offer opportunities for PDDL modelling and for translating natural language tasks into symbolic goals. An article under review titled “Leveraging Pre-trained Large Language Models to Construct and Utilize World Models for Model-based Task Planning” explores this further. See this [interaction with ChatGPT](https://chat.openai.com/share/c0cd35b5-dfdb-4320-972d-a9188d77f631) for examples.

- [AKFM22] Asai _et al_. Classical planning in deep latent space. J. Artif. Intell. Res., 74:1599–1686, 2022.
- [AFP+18] Arora _et al_._A review of learning planning action models_. The Knowledge Engineering Review, 33(e20):1–25, 2018.
- [MPSK22] Muise _et al_. PLANUTILS: Bringing Planning to the Masses. Demo track of ICAPS 2022. [PDF](https://icaps22.icaps-conference.org/demos/ICAPS_2022_paper_377.pdf)

---

### Decision · Program_Chairs · 2023-09-21

**Decision:**

Accept (poster)

**Comment:**

This paper leverages symbolic planners for effective egocentric planning over long-horizon task of following a instruction (in natural language). The authors demonstrate the efficacy of their simple baseline by achieving high performance on well-studied ALFRED benchmark in AI2THOR simulator. This authors do utilize an initial budget of 500 steps to explore/memorize the embodied agent's surroundings, which is represent as a semantic graph. The graph is transformed to a Planning Domain Definition Language (PDDL) formulation, making it amenable to classical planning approaches, which is done via an iterative exploration and replanning approach.

The reviewers are largely positive about the work and most of them concur that the paper has (1) a simple yet strong baseline for the standardized long-horizon planning benchmark of ALFRED, (2) effective use of symbolic planners, adding a novel take to end-to-end baselines in this research area, (3) good generalization, especially, in unseen scenes, and (4) good and clear presentation & writing.

The concerns around (1) difficulty of obtaining a symbolic model, (2) restrictive empirical evidence, i.e., only on ALFRED, and (3) need for 500 steps as initial exploration are drawbacks that were also shared by the reviewers. Efforts towards stating them clearly and upfront will help. Adding ablations that relax the aforementioned assumptions, could help the community to fairly benchmark EPA (or the most appropriate variation of it) in future works.

The AC concurs with the overall positivity of the reviewers and their appreciation for this alternative take at tackling long-horizon interactive planning in embodied environments and effective generalization.